# Prognostic parameterization of cloud ice with a single category in the aerosol-climate model ECHAM(v6.3.0)-HAM(v2.3)

Remo Dietlicher[1], David Neubauer[1], and Ulrike Lohmann[1]

[1]Institute for Atmospheric and Climate Science, ETH Zürich, Universitätstrasse 16, 8092 Zürich, Switzerland.

*Correspondence to:* Remo Dietlicher (remo.dietlicher@env.ethz.ch)

**Abstract.** A new scheme for stratiform cloud microphysics has been implemented in the ECHAM6-HAM2 general circulation model. It features a widely used description of cloud water with two categories for cloud droplets and rain drops. The unique aspect of the new scheme is the break with the traditional approach to describe cloud ice analogously. Here we parameterize cloud ice by a single category that predicts bulk particle properties (P3). This method has already been applied in a regional
model and most recently also in the climate model CAM5. A single cloud ice category does not rely on heuristic conversion rates from one category to another. Therefore, it is conceptually easier and closer to first principles.

This work shows that a single category is a viable approach to describe cloud ice in climate models. Prognostic representation of sedimentation is achieved by a nested approach for sub-stepping the cloud microphysics scheme. This yields good results in terms of accuracy and performance as compared to simulations with high temporal resolution. Furthermore, the new scheme
allows for a competition between various cloud processes and is thus able to unbiasedly represent the ice formation pathway from nucleation over growth by vapor deposition and collisions to sedimentation.

Specific aspects of the P3 method are evaluated. We could not produce a purely stratiform cloud where rime growth dominates growth by vapor deposition and conclude that the lack of appropriate conditions renders the prognostic parameters associated with the rime properties unnecessary. Limitations inherent to a single category are examined.

## 1   Introduction

Clouds are a major source of uncertainty in current climate projections as assessed by the last IPCC report (Stocker et al., 2013). Apart from synoptic-scale low pressure systems, clouds are not resolved by the coarse spatial resolution used in climate models, which necessitates a transfer from grid-box mean model-states over the sub-grid distribution of humidity down to the microphysical properties of clouds. The circumstances require heuristic methods to represent the average response of clouds
to natural and anthropogenic forcing.

Over the last 50 years the level of sophistication of the transfer methods from resolved to parameterized scales has steadily increased. Kessler (1969) built a scheme based on a system of continuity equations for vapor, cloud and precipitation, which assumed that clouds form as soon as grid-box mean supersaturation is established and precipitate proportionally to their mass. This idea was refined by the work of Sundqvist (1978) to account for sub-grid cloudiness by assuming an inhomogeneous
distribution of moisture within a model grid-box. Later on, polydisperse cloud droplets were represented (Beheng, 1994)

which was the first step towards the nowadays common transfer from grid-box mean quantities down to the particle scales by the assumption of particle size distributions in two moment schemes. Since then, a multitude of studies document the progress in both the representation of sub-grid clouds (e.g. Tompkins (2002)) and the extension of schemes based on particle size distributions to ice (e.g. Seifert and Beheng (2006)).

Li et al. (2012) assessed the ability of climate models to represent the amount of ice in clouds from the last generation of climate models. They found that the globally averaged, annual mean ice water path differs by a factor of 2 to 10 among the selected models used in the fifth model intercomparison project CMIP5. While all of the models in their study are in radiative balance, they do so at the cost of a wide variety of cloud ice contents due to the large uncertainty in their radiative properties. At the same time, new studies (Tan and Storelvmo, 2016) suggest that as the phase of a cloud is decisive for its radiative properties,

it strongly impacts the equilibrium climate sensitivity (ECS) (Tan et al., 2016). Satellite observations show furthermore that the occurrence of ice and mixed-phase clouds is tightly linked to precipitation fields (Muelmenstaedt et al., 2015; Field and Heymsfield, 2015) which further reinforces the importance of accurately representing cloud ice in models.

    The response of climate, and in particular clouds, to a warming world induced by increasing carbon dioxide emissions is a highly discussed topic in the climate research community (Hope, 2015; Bony et al., 2015; Stevens et al., 2016; Tan et al.,

2016; Schneider et al., 2017). Following Tan et al. (2016) we focus on improving the representation of the supercooled liquid fraction in climate models and hence cloud ice in general. As has been laid out by the study of Li et al. (2012) there is a lot of room for improvement in this area.

    Many climate models represent ice by predefining categories for a given particle characteristic, such as ice crystals, planar snow flakes or dense and spherical graupel and hail particles (Seifert and Beheng, 2006). Widely used categories for ice

particles in models are in-cloud ice and falling snow. With the coarse resolution employed in climate models, these categories serve to distinguish between cloud ice and precipitation. This approach is motivated by the analogous treatment for cloud liquid water where cloud droplets are separated from rain drops. However, unlike for liquid water where there is a clear scale separation between condensational growth and growth by collision and coalescence, the criteria to divide ice into an in-cloud category and a precipitating category is not well defined. This classification therefore differs from model to model and, being

weakly constrained, the associated conversion rates are often used as tuning parameters. The conversion from cloud ice to snow is usually based on a threshold size for snow. Some models cut off the particle size distribution at a given threshold (Morrison and Gettelman, 2008), others use it together with ice growth rates to calculate the time needed to grow particles to the threshold size (Murakami, 1990). Due to this heuristic partitioning, cloud ice parameterizations are associated with a large uncertainty.

    New studies (Morrison and Milbrandt, 2015; Jensen and Harrington, 2015) introduce techniques to describe cloud ice in a

more continuous fashion. Contrary to the common approach of representing ice as a composition of different particle types, they suggest to use a single category whose properties adjust smoothly to cloud conditions and formation history. This eliminates the need to parameterize weakly constrained conversion processes among categories.

    Describing a hydrometeor species with a single category implies that the entire category, including a potentially fast-falling part, has to be treated prognostically. Since prognostic precipitation categories are becoming more and more popular in multi-

category schemes as well (Gettelman and Morrison, 2015; Sant et al., 2015), many approaches exist to locally increase time

resolution in order to achieve numeric stability. Here we present a nested sub-stepping approach in the ECHAM6-HAM2 general circulation model (GCM) microphysics scheme.

In this study we focus on the pathways subsequent to ice initiation by a more physically-based description of cloud ice but acknowledge the importance of ongoing research to understand freezing mechanisms (Welti et al., 2014; Ickes et al., 2015; Marcolli, 2017) and resulting parameterization development (Phillips et al., 2013; Ickes et al., 2017). We will use the microphysical properties of ice described in Morrison and Milbrandt (2015) (hereafter MM15), also known as the predicted bulk particle properties (P3), and embed them in the ECHAM6-HAM2 cloud microphysics scheme.

The P3 method has originally been implemented in the regional model Weather Research and Forecasting Model (WRF). More recently, it has also been included in the global context of the Community Atmosphere Model 5 (CAM5) (Eidhammer et al., 2017). They focus on the description of the particle properties within P3 and the empirical parameter choices therein and discard the effects of riming on the particle properties a priori. Our manuscript documents the transition from diagnostic snow to a prognostic single category in ECHAM6-HAM2 on a technical level. We use the full P3 method, including the rime properties, and evaluate a cloud formation scenario that is consistent with the forcing provided by a GCM and might allow for significant rime formation. We show idealized mixed-phase cloud simulations to better understand the cloud formation and glaciation process with the new scheme. The intuition gained from this exercise then helps interpreting model output where usually only temporal and global averages are available and the information on individual clouds is lost.

A short summary of the P3 method is given in Sect. 2. As this new approach leads to a revision of the existing cloud microphysics scheme, the entire scheme is described in Sect. 3. Numerical challenges associated with the prognostic treatment of sedimenting ice are addressed in Sect. 4. Section 5 shows results of the new microphysics scheme in a 1-D single column setup. We highlight conceptual differences to the original two category scheme, potential simplifications in the context of a GCM and limitations inherent to a single category. Section 6 concludes this study by evaluating the feasibility and benefit from using a single category ice phase scheme in ECHAM6-HAM2.

## 2 Revision of the predicted bulk particle properties (P3)

This section serves as a review of the fundamental concepts of the P3 method presented in MM15. All the relevant aspects for the prediction of particle properties from the grid-box mean model state are covered.

Instead of the two categories for cloud ice and snow, the single category scheme uses four prognostic parameters describing a single category: the total ice mass mixing ratio $q_i$, total ice number concentration $N_i$, riming mass mixing ratio $q_{rim}$ and riming volume $b_{rim}$. For the particle size distribution the gamma distribution

$$N(D) = \frac{N_i \lambda^{\mu+1}}{\Gamma(\mu+1)} D^{\mu(\lambda)} e^{-\lambda D} \tag{1}$$

is chosen with the gamma function $\Gamma(\mu)$ and the three free parameters $N_i$, $\mu$ and $\lambda$. An empirical relationship between $\mu$ and $\lambda$ (Heymsfield, 2003) reduces the number of free parameters from three to two:

$$\mu = 0.076 \left(\frac{\lambda}{100}\right)^{0.8} - 2. \tag{2}$$

The parameter $\lambda$ is given in $\mathrm{m}^{-3}$. The scheme defines a regime dependent mass to size relationship

$$
m(D) = \begin{cases} \frac{\pi}{6}\rho_i D^3 & \text{for small spherical ice, } D < D_{th} \\ \alpha_{va} D^{\beta_{va}} & \text{for dendrites, } D_{th} < D < D_{gr} \\ \frac{\pi}{6}\rho_g D^3 & \text{for graupel, } D_{gr} < D < D_{cr} \\ \frac{\alpha_{va}}{1-F_r} D^{\beta_{va}} & \text{for partially rimed crystals, } D_{cr} < D. \end{cases}
\tag{3}
$$

The parameters $\alpha_{va}$ and $\beta_{va}$ that define the mass to size relation for dendrites are empirical constants derived for aggregates of unrimed bullets, columns and side-planes (Brown and Francis, 1995), $\rho_i$ is the density of ice and $\rho_g$ is the density of graupel.

The rime fraction $F_r = q_{rim}/q_i$ is predicted by the model. The scheme needs to predict four unknown parameters; two for the mass to size relationship and two for the particle size distribution. The mass to size relationship is defined by the two transition sizes $D_{gr}$ separating dendrites from graupel and $D_{cr}$ separating graupel from partially rimed particles, see Fig. 1. The particle size distribution is defined by the total number of particles $N_i$ and one of the gamma distribution shape parameters $\mu$ or $\lambda$ through the use of Eq. (2).

In the following it will be explained how those four unknown parameters are calculated from grid-box mean state defined by the four prognostic parameters: the total ice mass $q_i$, the rimed ice mass $q_{rim}$ the total ice number $N_i$ and the rimed ice density $B_{rim}$. The threshold sizes $D_{gr}$ and $D_{cr}$ are defined by requiring that the mass to size relationship increases monotonically with increasing particle diameter. This yields the three equations

$$
\frac{\pi}{6}\rho_i D^3 = \alpha_{va} D^{\beta_{va}} \Rightarrow D_{th} = \left(\frac{\pi\rho_i}{6\alpha_{va}}\right)^{\frac{1}{\beta_{va}-3}}
$$

$$
\alpha_{va} D^{\beta_{va}} = \frac{\pi}{6}\rho_g D^3 \Rightarrow D_{gr} = \left(\frac{6\alpha_{va}}{\pi\rho_g}\right)^{\frac{1}{3-\beta_{va}}}
$$

$$
\frac{\pi}{6}\rho_g D^3 = \frac{\alpha_{va}}{1-F_r} D^{\beta_{va}} \Rightarrow D_{cr} = \left(\frac{6\alpha_{va}}{(1-F_r)\pi\rho_g}\right)^{\frac{1}{3-\beta_{va}}} .
\tag{4}
$$

One directly sees that $D_{th}$ only depends on constants while the others depend on the graupel density $\rho_g$ and rime fraction $F_r$. To calculate the graupel density $\rho_g$, a system of equations needs to be solved. The value of $\rho_g$ depends on both the density of rimed ice $\rho_r$ that accumulated on a particle by wet growth and the density of the underlying dendritic structure $\rho_d$. It is thus calculated as the average of the two, weighted by the rime mass fraction:

$$
\rho_g = F_r \rho_r + (1-F_r)\rho_d
\tag{5}
$$

The density of the underlying dendrite in the rimed regime (illustrated by the dashed-dotted line in Fig. 1) in turn is calculated as a mass-weighted average over the rime regime ($D_{gr} < D < D_{cr}$):

$$
\rho_d = \frac{6\alpha_{va}(D_{cr}^{\beta_{va}-2} - D_{gr}^{\beta_{va}-2})}{\pi(\beta_{va}-2)(D_{cr}-D_{gr})} .
\tag{6}
$$

Equation (5) and Eq. (6) can be solved iteratively for $\rho_g$ through the use of Eq. (4). At this point, the mass to size relation $m(D)$ is completely defined. What remains is the shape parameter for the size distribution. For this, the integral equation

$$\frac{q_i}{N_i} = \int\limits_0^\infty m(D) \frac{\lambda^{\mu+1}}{\Gamma(\mu+1)} D^\mu e^{-\lambda D} dD \tag{7}$$

is solved for $\lambda$. Once all the parameters for $N(D)$ and $m(D)$ are determined from the predicted model parameters, any size-dependant process rate can be integrated offline and read back from a lookup table. This is necessary since the iterative process for finding $\rho_g$, solving Eq. (7) for $\lambda$ (or $\mu$) as well as integrating process rates such as the self-collection of ice particles over the four ice habit regimes is computationally too expensive to be done online.

The weakly constrained parameters in the scheme are the $\alpha_{\mathrm{va}}$ and $\beta_{\mathrm{va}}$ parameters in the $m(D)$ relation as well as the parameters describing the projected area $A(D)$ which is essential for the microphysical process rate calculations. The sensitivity to the involved parameter choices in the global context of CAM5 is elaborated in a recent publication by Eidhammer et al. (2017).

For this study we implemented the lookup table closely following the original P3 scheme and the empirical constants used therein.

## 3 Description of the cloud microphysics scheme

We developed a new cloud microphysics scheme in the framework of the ECHAM6-HAM2 (echam6.3.0-ham2.3-moz1.0) (Zhang et al., 2012; Stevens et al., 2013). The original cloud microphysics scheme solves prognostic equations for the mass mixing ratios of cloud liquid and ice (Lohmann and Roeckner, 1996). Snow and rain are diagnosed from the cloud mass mixing ratios of the respective phase. Over the years, this scheme was expanded to improve the representation of microphysical processes by adding prognostic equations for the number concentrations for cloud droplets and ice crystals (Lohmann et al., 1999; Lohmann, 2002). Conversion rates involving the ice phase date back to Lin et al. (1983) and Murakami (1990). Our inclusion of a completely new approach to describe ice properties brought changes to many parameterizations and required a complete restructuring of the code to allow for temporal sub-stepping. The following will describe the implementation of the microphysics scheme with a single category ice phase.

The snow category has been removed and instead, in addition to ice mass and number, the riming mass and riming volume are introduced to make up the four moment single category ice described in MM15. Except for the restructuring of the code explained in Sect. 3.1, the two category description of the liquid phase did not change. With the goal of better representing the supercooled liquid fraction, we changed the way the Wegener-Bergeron-Findeisen (WBF) process was parameterized. The original scheme did not allow deposition and condensation to occur simultaneously but parameterized the WBF process based on whether or not the sub-grid scale updraft velocity and hence cooling rates caused super- or subsaturation w.r.t. liquid water (Korolev and Mazin, 2003).

## 3.1 Code structure

To better understand the model integration, consider a cloud parameter $\phi$. This represents any of the prognostic parameters used in this scheme, e.g. cloud ice $q_i$ or cloud liquid $q_c$. The model then solves the equation

$$\frac{\partial \phi}{\partial t} + \mathbf{u} \cdot \nabla \phi = \left.\frac{\partial \phi}{\partial t}\right|_{\text{micro}} + \left.\frac{\partial \phi}{\partial t}\right|_{\text{vdiff}} + \left.\frac{\partial \phi}{\partial t}\right|_{\text{convection}} \tag{8}$$

where the left hand-side represents the resolved advection and the right hand-side the unresolved processes that need to be parameterized. The tendencies due to the microphysics routine are summarized here by $\left.\partial\phi/\partial t\right|_{\text{micro}}$. The tendencies $\left.\partial\phi/\partial t\right|_{\text{vdiff}}$ and $\left.\partial\phi/\partial t\right|_{\text{convection}}$ are calculated by ECHAM6-HAM2's vertical diffusion and convection modules (Stevens et al., 2013). Cloud microphysics modules not only include phase changes and aggregation processes, but also the vertical advection of precipitation. For the prognostic treatment of precipitation, this aspect has strong implications on the accuracy and performance

of the scheme. To this end, we separate the cloud processes from the prognostic advection of cloud ice. As will be elaborated carefully in Sect. 4, this allows to employ a two-step reduction of the global model time-step to 1) sufficiently resolve the computationally heavy cloud process calculations and 2) to achieve numerical stability for the vertical advection of cloud ice. This separation is shown in Fig. 2 by the *local update* boxes separating the calculation of *cloud processes* and *sedimentation*. In terms of the arbitrary cloud parameter $\phi^n$ at time-step $n$ the workflow of a single microphysics sub-step can be expressed

as:

$$\phi' = \phi^n + \left.\frac{\partial \phi^n}{\partial t}\right|_{\text{cloud}} \Delta t,$$
$$\phi^{n+1} = \phi' + \left.\frac{\partial \phi'}{\partial t}\right|_{\text{sed}} \Delta t. \tag{9}$$

The entire scheme then consists out of $n_{\text{out}}$ iterations of Eq. (9) where the calculation of the sedimentation tendency is done iteratively as well. This iterative process is shown in Fig. 2. The cloud process tendency is given by

$$\left.\frac{\partial \phi}{\partial t}\right|_{\text{cloud}} = \left.\frac{\partial \phi}{\partial t}\right|_{\text{iaccl}} + \left.\frac{\partial \phi}{\partial t}\right|_{\text{islf}} + \left.\frac{\partial \phi}{\partial t}\right|_{\text{raccc}} + \left.\frac{\partial \phi}{\partial t}\right|_{\text{cautr}} + \left.\frac{\partial \phi}{\partial t}\right|_{\text{e/s}} + \left.\frac{\partial \phi}{\partial t}\right|_{\text{c/d}} + \left.\frac{\partial \phi}{\partial t}\right|_{\text{act}} + \left.\frac{\partial \phi}{\partial t}\right|_{\text{ci}} + \left.\frac{\partial \phi}{\partial t}\right|_{\text{mlt}} + \left.\frac{\partial \phi}{\partial t}\right|_{\text{frz}}. \tag{10}$$

The individual terms are discussed in detail below. The abbreviations stand for: *iaccl* accretion of liquid by ice, *islf* self-collection of ice, *cautr* and *raccc* autoconversion and accretion of cloud droplets to rain, *e/s* below-cloud sublimation of rain and sedimenting ice, *c/d* cloud formation and dissipation in response to large-scale forcing, *act* activation of aerosol particles to cloud droplets, *ci* nucleation and deposition in cirrus clouds allowing supersaturation with respect to ice, *mlt* melting of ice, *frz* homogenous and heterogeneous freezing of cloud droplets.

Equation (10) is in strong contrast to the original scheme. Due to the long time-step of the global model, cloud processes have been calculated sequentially. This allowed to represent a full cloud life-cycle within one time-step; from condensation/deposition over collisions and freezing/melting to evaporation/sublimation or precipitation formation. With the sub-stepping introduced to resolve the vertical advection of cloud ice, the new scheme also resolves the life-cycle of rather short-lived clouds and hence does not need to introduce a specific order in which the cloud processes occur.

In practice, the tendencies for cirrus nucleation and deposition $\partial\phi/\partial t\big|_{\mathrm{ci}}$ and cloud droplet activation $\partial\phi/\partial t\big|_{\mathrm{act}}$ are currently computed before the outer loop for two reasons. Both parameterizations are based on the time rate of change of supersaturation within an adiabatic parcel ascent. Particle formation depends on the maximal supersaturation that can be reached before condensational/depositional growth quickly depletes all supersaturation established by cooling. Such parameterizations are designed for global models where the time-step is large enough, such that the entire process from parcel ascent to particle formation and subsequent depletion of supersaturation takes place within a single time-step. With a reduced time-step, this assumption does no longer hold. Furthermore, ECHAM6-HAM2 is designed in a modular way. Therefore, aerosol-related particle formation is not calculated within the cloud microphysics module but as part of the aerosol module HAM2 and a separate cirrus module. This approach allows to choose the appropriate scheme for different applications and decouples development of different modules by specifying a coupling interface.

In the following subsections the parameterizations used to calculate the individual terms in Eq. (10) are presented.

## 3.2 Cloud processes

### 3.2.1 Cloud formation and dissipation

Condensation and deposition can occur before grid-box mean supersaturation is established. The formation and dissipation of a cloud depends on the convergence and divergence of specific humidity and temperature (Sundqvist et al., 1989). The fractional cloud cover $b$ is related to the relative humidity RH:

$$b = 1 - \sqrt{1 - \frac{\mathrm{RH} - \mathrm{RH}_c}{1 - \mathrm{RH}_c}} \tag{11}$$

where $\mathrm{RH}_c$ is a threshold grid-box mean relative humidity that has to be exceeded for cloud formation to be initiated. The previous microphysics scheme used a threshold ice mass mixing ratio to decide whether to use the relative humidity w.r.t. ice or liquid. However, this approach handles glaciation of a cloud poorly and leads to a sudden increase in cloud cover once the threshold is exceeded. To circumvent this problem, we introduce a saturation specific humidity

$$q_s = f q_{s,l} + (1 - f) q_{s,i} \tag{12}$$

in the mixed-phase cloud regime, i.e. the temperature range from $-35\,^\circ\mathrm{C}$ to $0\,^\circ\mathrm{C}$. The parameter $f$ is a weighting function with $f(-35\,^\circ\mathrm{C}) = 0$ and $f(0\,^\circ\mathrm{C}) = 1$ and $q_{s,l/i}$ are the saturation specific humidities over liquid and ice. With that, an interpolated relative humidity $\mathrm{RH}^* = q/q_s$ is inserted into equation Eq. (11) to calculate the cloud cover.

For all microphysics processes, $b$ is used to calculate the in-cloud values, e.g. $q_i = \bar{q}_i/b$ where $q_i$ is the variable used for in-cloud processes and $\bar{q}_i$ is the grid-box mean value.

Sedimenting ice and rain, which are allowed to fall into cloud free layers where $\mathrm{RH}^* < \mathrm{RH}_c$, use a sedimentation cover based on the cloud cover of the precipitating cloud. The sedimentation cover $b_{sed}$ is simply diagnosed as the cloud cover at the base of the next cloud above.

The water mass $Q$ that is available for condensation/deposition (or required to evaporate/sublimate) is given by

$$Q = -b(\Delta q_f - \Delta q_s) \tag{13}$$

where $\Delta q_f$ is the moisture convergence in the grid-box by the resolved transport and $\Delta q_s$ is the change in saturation specific humidity due to heat advection which includes the change given by the Clausius-Clapeyron equation as well as the temperature dependence of the weighting function $f(T)$.

We follow the approach of Morrison and Gettelman (2008) to directly include the WBF process in mixed-phase clouds and calculate the mass of water that is able to deposit on the existing ice crystals (Lohmann et al., 2016)

$$A = \Delta t N_i \alpha_m f_v \frac{4\pi C (\text{RH}_i - 1)}{F_k^i + F_d^i} \tag{14}$$

where $\alpha_m = 0.5$ is the probability of a water vapor molecule to successfully be incorporated into an ice crystal, $\Delta t$ is the model time-step, $C$ is the diameter $D$ dependant capacitance of the ice particle ($C = D$ for spherical graupel and $C = 0.48D$ for dendritic particles). The parameters $F_k^i$ and $F_d^i$ are thermodynamic parameters depending only on temperature. The parameter $f_v$ is the ventilation coefficient given by a parameterization from Thorpe and Mason (1966) for plate-like ice crystals

$$f_v = 0.65 + 0.44 N_{sc}^{1/3} N_{re}^{1/2}, \tag{15}$$

with the Schmidt number $N_{sc}$ and the Reynolds number $N_{re}$. Since both the Reynolds number $N_{re}$ and the capacitance $C$ are size dependant, the respective summands are integrated offline and read back from the lookup table. This formulation is identical to the original P3 scheme. For the relative humidity, it is assumed that $RH_i = q_{s,l}/q_{s,i}$ i.e. that the cloudy portion of the grid-box is at water saturation as long as liquid water is present in mixed-phase clouds. Additionally, it is assumed that ice crystal growth is prioritized over cloud droplet growth. With those two assumptions, the following rules determine the condensation and deposition in mixed-phase clouds.

In a cloud forming environment ($Q > 0$), the mass potentially available for deposition is the sum of the excess water vapor $Q$ and the liquid water $q_c$. If $Q < A$ the missing water is taken from the liquid phase and thus represents the WBF process. Otherwise both cloud droplets and ice crystals grow. In the case of a dissipating cloud ($Q < 0$), cloud droplets evaporate first and only if $q_c < Q$ ice crystals sublimate.

The growth and dissipation of pure ice clouds in the mixed-phase regime follows that dictated by Eq. (13) to be consistent with the cloud fraction in Eq. (11). The formation of cirrus clouds is handled separately and are discussed in the next sub-section.

### 3.2.2 Cirrus clouds

Homogeneous freezing of solution droplets in cirrus clouds is considered. Starting at around $140\,\%$ RH with respect to ice, it is evident that the in-cloud deposition discussed in the previous sub-section is not suited to represent such clouds as it does not allow supersaturation by design. To capture this effect we allow supersaturation with respect to ice and parametrize cirrus clouds by the scheme described in Kärcher and Lohmann (2002). It is based on sub-grid updraft velocity inferred from

the turbulent kinetic energy (TKE) to obtain a more physical sub-grid distribution of saturation values. Vapor deposition is calculated explicitly based on the supersaturation w.r.t. ice. Cirrus clouds dissipate the same way as mixed-phase and liquid clouds.

This scheme has two main limitations. Firstly, it does not include pre-existing ice crystals that compete for available humidity with the newly formed ones. Secondly, the use of the TKE to infer sub-grid updraft velocities is debatable. A study by Joos et al. (2008) showed that this formulation in ECHAM5 did not reproduce the observed updraft velocities and better agreement with observation was reached by including orographic gravity waves. A recent study with CAM5 reached better agreement with observations by only using the large-scale updraft velocity (Zhou et al., 2016). Improving the representation of cirrus clouds is work in progress in our group.

### 3.2.3 Aerosol activation

The model ECHAM6-HAM2 used in this study is equipped with the online aerosol model HAM version 2 (Zhang et al., 2012). Number concentrations and mass mixing ratios of 5 aerosol species (sulfate, sea salt, mineral dust, black carbon and organic carbon) are calculated with the aerosol module HAM. Aerosol activation is calculated according to Abdul-Razzak et al. (1998) and Abdul-Razzak and Ghan (2000) considering the aerosol particle size and chemical composition. The transition from grid-box mean to the physically relevant sub-grid formulation is done according to the sub-grid updraft velocity. We apply a correction to cloud droplet number concentrations if the mass-weighted mean droplet size is unphysically large because aerosol activation was too weak. For that, we adjust the number concentration such that a volume mean droplet radius of $25\,\mu\text{m}$ is not exceeded.

### 3.2.4 Freezing of cloud droplets

The freezing capabilities of black carbon and mineral dust are calculated according to the parameterization developed by Lohmann and Diehl (2006). It accounts for both contact freezing of mineral dust and immersion freezing of black carbon and mineral dust in stratiform mixed-phase clouds.

### 3.2.5 Liquid-Ice interactions

Collisions between cloud droplets and ice crystals are calculated based on the ice particle's projected area and fall speed. These properties are part of the new single category description of ice and further described in MM15. The current diagnostic treatment of rain does not allow to calculate rain drop collection by ice. We assume that this process can be neglected and riming will be dominated by cloud droplets colliding with ice particles. This is equivalent to the original microphysics scheme in ECHAM6-HAM2.

### 3.2.6 Ice particle self-collection

With the size distribution and projected area to diameter relation intrinsic to the single category scheme we are able to numerically integrate the collection kernel. The resulting process rates are stored and read from lookup tables. This replaces the aggregation parameterization employed by the original scheme to calculate the efficiency with which ice crystals collide to form snow.

### 3.2.7 Below cloud evaporation/sublimation

The evaporation of rain is calculated according to Rotstayn (1997). For ice we use the same formulation for below cloud sublimation as employed for deposition by moisture convergence (Eq. (14)) where we use the grid-box mean subsaturation with respect to ice. This implies that we neglect a potential subgrid distribution of humidity in completely cloud-free grid-boxes.

### 3.2.8 Melting of ice

Melting is calculated based on Mason (1958). It combines terms for condensation, diffusion and riming in a heat-budget equation. Here only condensation and diffusion are considered:

$$\frac{\partial q_i}{\partial t}\Big|_{\mathrm{mlt}} = -\frac{2\pi D[K(T-T_0)+\rho_{air}\psi L_v(q_v - q_s|_{t=T_0})]f_v}{L_f} \tag{16}$$

where $L_{v/f}$ is the latent heat of vaporization and fusion respectively, $K$ is the thermal conductivity of air, $\rho_{air}$ is the density of air and $\psi$ is the water vapor diffusivity. The melted water is added to the cloud water mass within the cloud and to the rain water mass below the cloud.

### 3.2.9 Cloud droplet autoconversion and accretion by rain

Warm phase processes are adapted from the original microphysics scheme in ECHAM6-HAM2 to minimize differences and enhance comparability. The sedimentation of liquid water is diagnosed by a separate category for rain that is assumed to fall through the whole column within one single global time-step. Rain is formed by autoconversion and increased by accretion. Autoconversion from cloud water to rain is calculated from the cloud liquid mass mixing ratio $q_c$ and the number concentration of cloud droplets $N_c$ following the empirical relation (Khairoutdinov and Kogan, 2000):

$$\frac{\partial q_r}{\partial t}\Big|_{\mathrm{aut}} = 1350 q_c^{2.47} N_c^{-1.79} \tag{17}$$

where $q_r$ is the rain mass mixing ratio. Rain falling from above is also able to grow by accretion of cloud droplets following:

$$\frac{\partial q_r}{\partial t}\Big|_{\mathrm{acc}} = 3.7 q_c q_r \tag{18}$$

The rain flux is then given by (Stevens et al., 2013)

$$P_{\mathrm{rain}} = \frac{1}{g}\int_0^p (S_{\mathrm{aut}} + S_{\mathrm{acc}} + S_{\mathrm{mlt}} - S_{\mathrm{evp}})dp. \tag{19}$$

for pressure $p$ and the source and sink terms of autoconversion from cloud droplets to rain $S_{\mathrm{aut}} = \frac{\partial q_r}{\partial t}\big|_{\mathrm{aut}}$, accretion of cloud droplets by rain $S_{\mathrm{acc}} = \frac{\partial q_r}{\partial t}\big|_{\mathrm{acc}}$, melting of ice $S_{\mathrm{mlt}}$ and evaporation of rain $S_{\mathrm{evp}}$. Given the precipitation velocity of rain, the rain mass mixing ratio $q_r$ used for the accretion rate can be calculated from the rain flux $P_{\mathrm{rain}}$.

### 3.3 Sedimentation

#### 3.3.1 Falling ice

Sedimentation of ice is calculated prognostically according to MM15. The rate of change due to sedimentation is deduced from the number-weighted mean ($v_n$) and mass-weighted mean ($v_m$) fall speeds

$$v_n = \frac{\int_0^\infty v(D)N(D)dD}{\int_0^\infty N(D)dD} \tag{20}$$

$$v_m = \frac{\int_0^\infty v(D)m(D)N(D)dD}{\int_0^\infty m(D)N(D)dD}. \tag{21}$$

The fall speeds are computed offline and are read back from lookup tables.

The rate of change due to sedimentation is given by a one-dimensional advection equation:

$$\frac{\partial \phi}{\partial t} + v_{m/n}\frac{\partial \phi}{\partial z} = 0 \tag{22}$$

where $\phi$ represents the ice moments: number- $N_i$, ice- $q_i$, rimed ice- $q_{rim}$ and volume mixing ratio $B_{rim}$. The number mixing ratio sediments according to the number-weighted mean fall speed $v_n$, the other three according to the mass-weighted mean fall speed $v_m$.

Given the long time-step of a global model, large errors will arise in the vertical advection of cloud ice. Therefore substepping was applied to the relevant part. This will be further explained in Sect. 4.

## 4 Treating prognostic sedimentation efficiently

The standard version of ECHAM6-HAM2 diagnoses precipitation assuming it reaches the surface within one global model time-step. Treating sedimentation prognostically requires much smaller time-steps to resolve the vertical motion of hydrometeors. We therefore introduce temporal sub-stepping in the microphysics and sedimentation calculations in order to achieve numerical stability and keep numerical errors small.

The perfect integration method to solve the vertical advection equation for sedimenting ice (Eq. (22)) should be non-dispersive, unconditionally stable and be able to deal with sharp wave-fronts usually encountered at cloud base and cloud top. Unfortunately, this integration method does not exist. Here we use the upstream version of an explicit Euler method which leads to the following discretization of Eq. (22):

$$\phi_k^{n+1} = \phi_k^n + \Delta t\left(\frac{v_{k-1}^n \phi_{k-1}^n}{\Delta z_{k-1}} - \frac{v_k^n \phi_k^n}{\Delta z_k}\right) = \phi_k^n + \alpha_{k-1}^n \phi_{k-1}^n - \alpha_k^n \phi_k^n. \tag{23}$$

Indices $n$ represent the $n^{th}$ time-step and indices $k$ represent the $k^{th}$ model level. We introduced the Courant-Friedrich-Lewy (CFL) number $\alpha(\Delta t) = v\Delta t/\Delta z$ because it is a useful quantity to assess the numerical stability of a method. It can be interpreted as the number of levels that are passed within a time interval $\Delta t$. This scheme is stable for $\alpha < 1$ and dispersive for large time steps. In the following we will present a method exploiting the sequential treatment of cloud processes and sedimentation to assure numerical stability and reasonable accuracy while reaching optimal model performance.

## 4.1  The optimization strategy

In this section we present an approach to find a compromise between computational efficiency and model accuracy. The goal is to increase model efficiency by distributing the workload between the two parts shown in Fig. 2; the computationally expensive outer loop with $n_{\mathrm{out}}$ iterations calculating both cloud process rates and sedimentation, and the computationally cheap inner loop with $n_{\mathrm{in}}$ iterations calculating only the sedimentation of cloud ice. Sedimentation is calculated roughly 100 times faster than the cloud processes. Since the loops are nested, the number of calls to the sedimentation calculation will be $n_{\mathrm{tot}} = n_{\mathrm{out}} \times n_{\mathrm{in}}$, reducing the time-step in the sedimentation calculation and cloud processes to $\Delta t_{\mathrm{in}} = \Delta t/n_{\mathrm{tot}}$ and $\Delta t_{\mathrm{out}} = \Delta t/n_{\mathrm{out}}$ respectively. To achieve numerical stability, we calculate the required number of sub-steps $n_{\mathrm{tot}}$ online to achieve $\alpha_k(\Delta t_{\mathrm{in}}) = v_k\Delta t_{\mathrm{in}}/\Delta z_k < 1$ for every level $k$. This requirement could be relaxed for an implicit scheme which is stable even for $\alpha > 1$. Since the outer loop is so much slower than the inner, the restriction to $\alpha < 1$ is not our main concern because it can easily be achieved by the inner loop. This leads to the following expression for $n_{\mathrm{tot}}$ on $N$ model levels:

$$n_{\mathrm{tot}} = \max_{k \in 1,...,N} \alpha_k(\Delta t). \tag{24}$$

The calculation of the process rates is not subject to the same restrictions as sedimentation. However, if we only use the inner loop ($n_{\mathrm{out}} = 1$) to achieve numerical stability, we will not be able to represent the process rates acting on falling particles and impair model accuracy. On the other hand, if we set $n_{\mathrm{in}} = 1$, we will have to achieve numerical stability solely by the expensive outer loop and impair computational efficiency.

We found a compromise between the two extremes by considering the trajectory of a falling ice crystals. For model accuracy it is important, that the cloud processes are calculated every time the ice crystals reach a new model level and are thus exposed to a new environment. This is equivalent to the requirement $\alpha(\Delta t_{\mathrm{out}}) < 1$. However, if fall speeds are very high and/or the level is very thin, the total rate of change of cloud ice will be dominated by sedimentation. Processes like sublimation and melting will not have enough time to significantly change the cloud ice content on those levels. Therefore, we neglect the last part of the trajectory where the ice only spends a fraction of a global model time-step and calculate the number of outer iterations:

$$n_{\mathrm{out}} = \alpha_{\max} = \max_{k \in I_{\mathrm{out}}(x)} \alpha_k(\Delta t) \tag{25}$$

for the set $I_{\mathrm{out}}(x) = \{k \mid k \in \{1,...,N\} \wedge \Sigma_{i=0}^{k}\tau_k > x\}$ for the time spent on level $k$ $\tau_k$, a specified threshold time $x$ and a model with $N$ levels. Numerical stability is then achieved on all levels by using $n_{\mathrm{in}} = n_{\mathrm{tot}}/n_{\mathrm{out}}$ inner iterations.

We illustrate our approach using the test case properly described in the next section. For the purpose of this section, the exact model setup is not important. Given the distribution of cloud ice in Fig. 3a and fall speed in Fig. 3b we can calculate the time

that cloud ice will spend in each level. Since the thickness of model levels varies from $90\,\text{m}$ at the surface to $700\,\text{m}$ at $8\,\text{km}$ height and since gravitational size-sorting and self-collection lead to the fastest fall speeds close to the surface, $\alpha$ varies from 57 at the surface to 1 at $8\,\text{km}$ (Fig. 3c). At the same time the accumulated time spent below a certain height is only a fraction of a global time-step (Fig. 3b).

The colored bars in Fig. 3c show different choices for the threshold time $x$ in units of the global model time-step $\Delta t$. Its value ranges from 0 (orange bar), meaning that the entire trajectory is considered, to 1 (brown bar), neglecting the part of the trajectory where ice does not spend at least one global model time-step.

    For the following estimation of the model error we chose the strategy corresponding to the purple bar $I_{\text{out}}(0.2)$. We neglect the last part of the trajectory where cloud ice spends less than $20\,\%$ of the global model time-step. This choice represents a

compromise between performance and accuracy will be further elaborated below.

## 4.2   Estimating the model error

To demonstrate the power of the sub-stepping described above, we employ a simple single column setup with 31 vertical levels. At one single model level at around $8\,\text{km}$ height a constant ice source term with specified tendencies for all 4 ice moments is prescribed at every time-step. The source terms are chosen such that the ice particles reach very high fall speeds and thus sub-

stepping is fundamentally important. The forcing is representative for hail formation and thus an extreme case that is probably not produced very often by the global model. However, it highlights the need for sub-stepping while the general conclusions are also true for smaller or dendritic particles.

    We prescribe the tendencies for cloud ice with $\partial q_i/\partial t = \partial q_{rim}/\partial t = 5 \times 10^{-3}\,\text{g}\,\text{kg}^{-1}\,\text{s}^{-1}$, $\partial n_i/\partial t = 1 \times 10^{3}\,\text{kg}^{-1}\,\text{s}^{-1}$ and $\partial b_{rim}/\partial t = \partial q_{rim}/\partial t \rho_{rim}^{-1}$ where we set the rime density to $\rho_{rim} = 900\,\text{kg}\,\text{m}^{-3}$. This forcing implies a rime fraction $F_r = 1$.

The temperature profile is prescribed, constant in time and decreases linearly from $0\,°\text{C}$ at the ground to $-55\,°\text{C}$ at $8\,\text{km}$. Relative humidity is set to $50\,\%$ w.r.t. liquid water. The relevant microphysical processes are sedimentation, sublimation and self-collection. The resulting ice profile undergoes a build-up phase until it reaches an equilibrium such that the source term is balanced by the precipitation sink.

    Results from this idealized experiment are shown in Fig. 4 and Fig. 5. Since we are investigating numerical errors due to

insufficient time resolution, the high-resolution run (*T6*; black, dashed line) with a time-step of $6\,\text{s}$ is regarded as the truth in the following analysis. We did not change the vertical resolution, therefore CFL-numbers $\alpha(\Delta t = 6\,\text{s})$ are very small throughout the column and no sub-stepping needs to be applied in the *T6* case. With the large CFL-numbers for the global time-step, errors in the sedimentation calculation are huge. The simulation without sub-stepping (*NO*) shows that the large errors in the sedimentation calculation lead to a numerical deceleration of sedimentation. This in turn strongly delays surface precipitation

and leads to an accumulation of ice in the atmosphere, see Fig. 4 (orange lines). The opposite problem is encountered when only the inner loop is acting to reduce $\alpha$ (*IN*; purple line). While the error in the sedimentation calculation is reduced, the sequential treatment of cloud processes and sedimentation leads to an underestimation of sublimation and thus overestimation of surface precipitation.

The simulations *OUT* (green line) and *FL* (light blue line) reproduce the results of the high-resolution simulation *T6* much better. The two simulations only differ by the fact that the *FL* simulation further reduces $\alpha$ by the additional sub-stepping of the sedimentation calculation to achieve $\alpha < 1$ throughout the column by the inner loop. This difference is most pronounced in the low, thin levels where $\alpha$ can still be very large even if it is reduced by the outer loop already. The effect of this can be

seen by comparing the vertical profiles of the process rates of the two simulations in Fig. 5. In the lowest levels, the simulation *OUT* deviates from the reference *T6* which leads to an overall error in the surface precipitation and the cloud ice profile. The simulation *FL* reaches very good agreement with the reference at all levels by assuring $\alpha < 1$ and thus achieving numerical stability throughout the column.

## 4.3   Linking performance and accuracy

The test case presented above allows to use the optimization strategy to find a compromise between performance and accuracy. To assess performance, we measure the computation time of the cloud microphysics routine. Specifically the two parts *cloud processes* and *sedimentation* shown in Fig. 2. We then define the speed-up as the ratio $t_{\text{T6}}/t_{\text{SIM}}$ of the computation times $t$ of any simulation *SIM* and the reference high-resolution simulation. Figure 6a shows the relative errors in surface precipitation and ice water path at equilibrium together with the speed-up for each simulation. A new simulation *OUT100* is introduced to

provide a further benchmark. It uses only the outer loop to achieve $\alpha < 1$ throughout the column, i.e. puts all the work into the expensive cloud processes iteration. This is equivalent to the strategy illustrated by the orange bar in Fig. 3c.

Figure 6a illustrates that the optimization strategy works as expected and a speed-up of around 7 can be achieved by only considering part of the column to calculate the number of outer iterations. It also confirms the finding form the last section; using the inner loop is essential to reduce the error from more than $15\,\%$ in the *OUT* simulation to well below $5\,\%$ in the

*FL* simulation while keeping the speed-up almost identical. The simulation *OUT100* achieves an almost exact match with the high-resolution simulation. However, it comes at the cost of drastically reducing model performance. Thus, the benefit of the sequential treatment of cloud processes and nested sub-stepping becomes clear.

Figure 6b adds more depth to the optimization strategy. By considering a range of different threshold times $x$, we can choose to trade accuracy for performance. By setting a high threshold time, we can achieve a speed-up of up to $15$ if we accept the

larger error.

This analysis has been performed on a series of different test cases (varying particle size and density as well as varying temperature and relative humidity profiles; not shown), including one where ice melts to form rain. This last test is particularly interesting because the melting layer represents a sharp change of process rates from one level to the next. However, since the calculation of rain is largely independent of the sub-stepping (see the section below) and since melting is represented by a

finite, physically-based time scale, the optimization strategy did not lead to large errors. We chose to show a different test case here because we wanted to highlight the treatment of the sedimentation of cloud ice in the lowest levels where model levels are very thin.

While the values for the relative error vary roughly between $0\,\%$ and $5\,\%$ depending on the test case and threshold values, the overall correspondance of relative error and speed-up has shown to be a robust result of our optimization strategy.

## 4.4 Sub-stepping and the diagnostic treatment of rain

This section provides a closer look at the rain flux within the sub-stepping environment. A diagnostic treatment of precipitation is designed for very large time-steps where the vertical movement of rain drops cannot be resolved. Since the new scheme in principle allows to resolve falling hail particles, we are outside of the realm the rain flux scheme was originally designed for. Since this work focuses on the representation of cloud ice, we will not discuss potential improvements to the liquid phase that would benefit from the newly employed sub-stepping. The obvious improvement would be using prognostic rain as was done by Sant et al. (2013). However, since their approach was different in terms of treating the cloud droplet and rain drop size distribution, a merging of these 2 approaches is beyond the scope of this paper but will be envisioned in future. Sticking with the rain flux approach, it is important to rule out any systematic biases of rain production associated with the sub-stepping employed for cloud ice.

To estimate the sensitivity of rain production to the number of outer iterations, we use a similar setup as for the ice sedimentation: A single column simulation with an isothermal atmosphere at $20\,^{\circ}\mathrm{C}$ and a relative humidity of $100\,\%$ throughout the column. A humidity tendency of $5 \times 10^{-7}\,\mathrm{kg\,kg^{-1}\,s^{-1}}$ is applied to the model levels between 16 and 26 (corresponding to $900\,\mathrm{hPa}$ and $400\,\mathrm{hPa}$ in pressure levels). This forcing is representative of stratiform cloud formation in the global setup of the model and corresponds to a water column tendency of $2\,\mathrm{mm\,h^{-1}}$. For this experiment we fixed the cloud droplet number concentrations but vary their (constant) values from $50\,\mathrm{cm^{-3}}$ to $1000\,\mathrm{cm^{-3}}$ to represent clouds with stronger and weaker rain production rates.

The simulations are run for one day with the humidity forcing active throughout the whole simulation. Every simulation is run once with $n_{out} = 1$, i.e. without sub-stepping affecting rain production and once with $n_{out} = 100$, i.e. with a very large number of outer iterations. Figure 7 shows the vertically integrated rain production rate together with its constituents: rain enhancement by accretion of cloud droplets by rain drops and autoconversion of cloud droplets to rain drops.

The first row in Fig. 7 indicates that different numbers of sub-steps and thus different time-step lengths can lead to differences in the rain production rate. The simulation with 1 sub-step has a slightly delayed onset of precipitation and tends to overshoot the total rain production by up to $10\,\%$ before reaching an equilibrium. For the simulations with the highest number concentrations and therefore weakest rain production rates, both the delay in onset and relative strength of overshooting is less pronounced. Eventually, every simulation reaches an equilibrium where the humidity input is balanced by the rain sink. This external constraint leads to vanishing differences in equilibrium rain production rates for different numbers of sub-steps. The second and third rows show the constituents of the total rain production rate. These rates show that there is no compensation of errors by the accretion and autoconversion rates but rather that the differences are due to the overestimation of precipitation production by the linearized numerical integration method employed by the core model. The local update of $q_c$ and $N_c$ by sub-stepping Eq. (17) and Eq. (18) reduces the numerical error of the accretion and autoconversion rates and prevents the overshooting of precipitation formation that can be seen in the $n_{out} = 1$ simulations. This claim is backed up by a simulation with high temporal resolution which is almost identical to the simulation with 100 sub-steps and therefore not shown. We conclude that sub-stepping is beneficial for the representation of the rain flux but the effect is small.

Varying the number of outer iterations from 1 to 100 is an extreme case. In the global model setup the number of outer iterations ranges on average from 5 in the tropics to 25 in mid-latitudes. Since the model converges quickly with increasing number of outer iterations, the delayed onset and overshooting effects discussed here are an upper limit. We conclude that for our purpose a diagnostic scheme for liquid water is compatible with the prognostic treatment of cloud ice and no systematic biases are induced by the number of outer iterations used. This is important as the number of outer iterations is computed online and may vary between columns.

## 5 Simulations of an idealized mixed-phase cloud

To demonstrate the behavior of the new scheme we look at results from a more elaborate single column simulation than the ones used in Sect. 4. The setup is summarized by the initial and forcing profiles shown in Fig. 8. It allows for an isolated examination of the microphysics scheme by deactivating the convection, vertical diffusion and radiation parameterizations and allowing no surface evaporation. The forcing terms are chosen such that there are two cloudy regions: One in the cirrus and one in the mixed-phase regime (Fig. 8c). We run the simulations for $36\,\mathrm{h}$. The cirrus cloud forcing is applied from hours 3 to 4. In the mixed-phase cloud regime, the forcing is applied throughout the first $12\,\mathrm{h}$ (solid lines in Fig. 8c). After $12\,\mathrm{h}$ the humidity tendencies are set to zero for another $12\,\mathrm{h}$. Finally, tendencies equal in magnitude and duration but with opposite sign (dashed lines in Fig. 8c) are applied after $24\,\mathrm{h}$ such that the total water content is the same for simulation times $0\,\mathrm{h}$ and $36\,\mathrm{h}$. The temperature is kept constant throughout the simulation to compensate for latent heating by condensation/evaporation and deposition/sublimation. Since the vertical diffusion and convection parameterizations are turned off, this assures that the melting layer remains at the same level throughout the simulation which facilitates the interpretation of the results.

We prescribe mineral dust and sulfate aerosols which dominate heterogeneous freezing in mixed-phase conditions and homogeneous nucleation in cirrus clouds respectively. Cloud droplet number concentration is fixed at a constant value representative of marine clouds of $100\,\mathrm{cm}^{-3}$ since we are mainly interested in the evolution of cloud ice.

### 5.1 Comparison to the original scheme

The standard version of ECHAM6-HAM2 is equipped with a 2 moment scheme for both cloud liquid water and ice and diagnostic equations for snow and rain mass mixing ratios. For comparability, the new scheme can switch between calculating in-cloud and sedimentation tendencies based on the new single category and the original two category scheme. This way we are able to consider only the differences between the schemes that are due to the conversion of cloud ice to snow in the original scheme and the single category approach while all compatible cloud processes (vapor deposition, melting and freezing) are identical.

In the following we will present results from three different microphysics schemes, summarized in Table 1. They are presented in the order similar to the evolution within ECHAM-HAM. The *2 category scheme* treats ice and liquid water analogously by separating in-cloud and precipitation hydrometeors. While for liquid water this separation can be justified by the different scales on which growth by condensation and growth by collision-coalescence are efficient, the analogous argument

does not hold for cloud ice; a perfect dendrite can reach a significant fall speed. This deficiency has been solved by including the mass flux divergence scheme (Rotstayn, 1997) to allow the in-cloud part of the ice population to fall (Lohmann et al., 2008). Here we call this scheme the *2.5 category scheme* since the falling ice mass flux resembles a separate category. The problem with this is, that there is no physical distinction between falling ice and snow. Sublimation and melting are parameterized for

both precipitation categories, but only for snow collisions with liquid water is included. This leads to an artificial competition between falling ice and snow that, depending on the cloud forcing, forms precipitation hydrometeors that can further grow by riming or are limited to sublimation and melting. Furthermore, other growth mechanisms (i.e. self-collection and vapor deposition) are neglected for both precipitation categories. The *1 category scheme* is the logical successor in this line of development. By resolving the vertical advection of cloud ice, precipitation categories are no longer necessary. The spectrum of ice particles

is represented and one single set of cloud processes is parameterized for the entire ice hydrometeor population. With that, the conceptual problems of the previous schemes are solved. The cloud liquid and ice water contents of the three simulations in Fig. 9 highlight the differences between the schemes.

     In the *2 category scheme* simulation, cloud ice is not allowed to fall. In the single column model, ice crystals are therefore restricted to the level they formed in, which are the levels where heterogeneous freezing of cloud droplets takes place. Since

temperature decreases with increasing altitude, this process is most active at cloud top. As soon as a sufficiently large number of ice crystals has formed, their accumulated depositional growth is able to quickly deplete the co-existing liquid water. The large mass transfer from the liquid to the ice phase grows the ice crystals to a size where conversion to snow is efficient. Those snow particles partly deplete the liquid cloud below the freezing levels by riming and subsequently sublimate.

     The results from the *2.5 category scheme* simulation show how the situation changes when the ice crystals themselves are

allowed to fall. The ice crystals that formed at cloud top do not accumulate in the levels of formation but spread throughout the cloud. An exponential tail of the ice crystal mass flux continually falls out of the cloud where it sublimates. Due to the steady removal of cloud ice, growth by vapor deposition is delayed (orange lines in Fig. 10a and b). The snow production rate is weak because ice sediments out of the cloud before it can efficiently grow to the snow threshold size. Consequently, the riming rate is virtually zero since only collisions between snow flakes and cloud droplets are considered (purple line in Fig.

10c). The low riming and deposition rates lead to a mixed-phase cloud that is heavily dominated by liquid water (Fig. 10d). The challenge of treating the sedimentation of ice crystals diagnostically has been discussed in Rotstayn (1997). Our results support the hypothesis that a diagnostic scheme likely overestimates sedimentation.

     The *1 category scheme* does not rely on a seperate set of microphysical processes that are calculated only for diagnostic sedimentation categories like snow and falling ice in the *2.5 category scheme*. Thus, there can be a competition between

riming, vapor deposition and the removal by sedimentation. Residence times per level are resolved by the sub-stepping which allows to compute physical processes on a theoretical basis instead of empirical relationships employed for the mass flux calculation for ice and snow.

     These simulations nicely illustrate the theoretical considerations at the beginning of this section. The separation into multiple categories and the associated conversion rates strongly influence the resulting cloud structure, life-time and phase fraction. A

small snow threshold size leads to more snow and thus more riming, while for the threshold size of $100\,\mu m$ used in the *2.5*

*category scheme*, snow formation is almost completely inhibited by the steady removal of cloud ice by the mass flux divergence scheme. This sensitivity to a development choice that is not constrained by first principles is highly undesired. The problem is resolved by the prognostic, single category that does not have this degree of freedom and thus simulates the cloud in an objective, physically-based manner.

## 5.2 Predicting the rime fraction and density

The rime variables in the single category scheme, i.e. the rimed ice mass mixing ratio and the rimed ice volume mixing ratio, determine the density and shape of the particles with heavily rimed particles being spherical and weakly rimed particles having dendritic geometry. As a result, particles with high rime fractions have a smaller projected area and thus a higher fall speed than their weakly rimed counterparts of the same mass. Morrison and Milbrandt (2015) showed in a regional model that this adjustment of particle properties is crucial to correctly predict precipitation rates in a squall line simulation with strong convective updraft.

The ECHAM6-HAM2 GCM does not resolve those strong convective updrafts but parameterizes convection by the Tiedtke (1989) scheme. In its standard version, it employs very simplified microphysics which does not account for the co-existence of liquid water and ice. It is therefore questionable whether riming as such, and the resulting change in particle properties especially, is currently adequately represented in convective clouds.

From a purely stratiform cloud perspective, the effect of the particle properties on process rates is best illustrated by a seeder-feeder situation. Ice crystals form in the cirrus cloud and quickly grow to a few $100\,\mu m$ in diameter in the highly supersaturated environment. The largest crystals sediment quickly and subsequently interact with a supercooled liquid cloud below by riming and the WBF process. Whether riming or vapor deposition dominates strongly depends on the particle size. A few large particles will grow more strongly by riming while the growth of many small particles will be dominated by vapor deposition. Therefore, combining strong depositional growth in the cirrus regime with gravitational size-sorting provides the optimal conditions for rime growth within the supercooled liquid cloud. In the following, we will explore this special case and shed light on the particle properties within the P3 parameterization.

The boundary and forcing profiles for the simulation are the same as in the last section (Fig. 8) but with heterogeneous freezing in the mixed-phase cloud turned off. To investigate the sensitivity to the particle properties within the new scheme, we vary the effect that riming has on the mass to size relationship of the P3 scheme described in Sect. 2. Two simulations are done: The *rime properties* simulation uses the regular particle properties of the P3 scheme and the *dendritic properties* simulation neglects the effect of riming on the mass to size relationship. With the notation from Sect. 2 this can be expressed by setting the rime fraction $F_r = 0$ which results in $D_{gr} = D_{cr}$. The only remaining transition parameter then is $D_{th} \equiv const$, separating the small spherical ice regime from the dendrite regime.

Fig. 11 shows a summary of the process rates and column integrated water masses. We can see that neglecting the impact of riming on particle properties changes the thickness of the liquid cloud by roughly $10\,\%$. This can be attributed to the different riming and deposition rates in the two simulations which is ultimately a result of the slightly longer residence times within the mixed-phase cloud due to the smaller fall speeds in the *dendritic properties* simulation.

Fig. 12 shows the particle properties for the two simulations. From the process rates (Fig. 11b and c), we can see that rime growth only exceeds growth by vapor deposition at about 7 h after the simulation start and quickly diminishes after that. This is due to the gravitational size-sorting of the cirrus particles and the strong dependence on particle fall speed and diameter of the riming rate; large particles will reach the liquid cloud first and a significant amount of rimed ice forms upon impact with the cloud droplets. This is the time, where the two simulations differ most. While the particles from the *rime properties* simulation get slightly more spherical by riming and their fall speeds increase up to almost twofold, the particles from the *dendritic properties* simulation neglect the rime fraction for the fall velocity calculation. Therefore these particles keep their dendritic shape and fall more slowly which gives them more time to grow throughout the liquid cloud. The total ice particle density is largely dominated by the particle size. A fully rimed particle could reach a density of a several hundred $\mathrm{kg\,m^{-3}}$. However, the maximal rime fraction obtained here is roughly $60\%$. Therefore, the particles do not reach densities higher than $100\,\mathrm{kg\,m^{-3}}$.

The idealized, purely stratiform and turbulence-free simulations shown in this section do not produce the conditions necessary to form heavily rimed particles even though the setup has explicitly been designed to create an environment that favors rime growth over depositional growth. It is therefore questionable whether the global setup is able to provide the forcing necessary to form heavily rimed particles at all. While this is subject to further investigation, the simulation shown here suggests that for stratiform clouds, the computational cost to solve prognostic equations to predict the rime fraction and rime density could be saved.

## 5.3 Limitations of a single category

The single category scheme is able to represent a wide range of particle properties representing small in-cloud ice crystals, larger dendrites and fast falling structures like graupel and hail.

It is important to remember that the particle properties are parametrized by the particle size distribution, mass to size and mass to projected area relationships. Therefore a predefined relationship between the four prognostic ice parameters and the particle properties exists, which is layed out in Sect. 2. We would like to stress the fact, that it is the four predicted ice parameters (ice mass mixing ratio $q_i$, rimed ice mass mixing ratio $q_{rim}$ total ice number mixing ratio $N_i$ and rimed ice volume $B_{rim}$) that are prognostic and not the particle properties themselves.

This section provides a closer look at this pecularity of a single category scheme. We use a similar seeder-feeder simulation as in the last section but with a mixed-phase cloud instead of a pure liquid cloud as the feeder cloud. We will focus on three particular areas in the simulation, shown in Fig. 13a: 1) the mixed-phase cloud just before the cirrus particles impact, 2) the cirrus particles just above the mixed-phase cloud and 3) the resulting particles after impact.

From the corresponding particle size distributions in Fig. 13b we can see how a single category handles the addition of two ice masses with differing particle properties. The number concentration in the tail of the very large particles from the cirrus cloud (purple line) is lost by averaging even though it contributes almost $10\%$ to the total mass mixing ratio. This is because the total number concentration is heavily dominated by the mixed-phase cloud (orange line) with the cirrus cloud only making

up a small fraction of about $1\%$ thereof. Since the microphysical process rates are calculated from the resulting particle size distribution (green line), particle collisions and the sedimentation sink are likely underestimated.

A solution to this issue has been proposed by Milbrandt and Morrison (2016) by the use of multiple free categories. Ice of different origin will then be sorted according to its diagnosed properties and stored in separate variables, thus giving the properties themselves a prognostic flavour. However, adding prognostic variables for multiple free categories is computationally expensive. We argue that in the context of climate projections where we are interested in global and regional mean states of the atmosphere, the computational cost of this additional procedure likely outweighs the benefit from an improved representation of ice particle properties.

While a high-resolution model is able to produce different particle properties within a single cloud, the global model often represents a cloud with only a few grid-boxes. Therefore the regions where the use of multiple free categories could improve the model results are these where the seeder-feeder mechanism or convective anvils contribute significantly to the ice water path. How important these situations are in the global context in ECHAM6-HAM2 will have to be investigated in future studies. Then the benefit from multiple free categories can be revisited.

## 6  Conclusion

The single category scheme proposed by MM15 has been successfully implemented in the ECHAM6-HAM2 microphysics scheme. The structure of the original code has been reworked and the large-scale deposition, cloud cover and melting calculations have been adapted to comply with the prognostic ice category and a variable time-step. Numerical stability is achieved by sub-stepping the cloud microphysics and sedimentation routines with an attempt to keep computation time as low as possible by applying a nested sub-stepping approach. It has been shown that a compromise can be reached to allow reasonable accuracy and performance at the same time.

The new scheme is evaluated against its forerunner within an idealized mixed-phase cloud simulation. The sub-stepping introduced in the new scheme allows cloud ice sedimentation to be calculated prognostically and treats all cloud processes equally. This means, that falling ice is now subject to all the cloud processes, including all growth mechanisms (i.e. vapor deposition, self-collection and riming). The artificial competition between the two sedimentation categories that has been present in the original scheme, its implications for the process rates and hence the cloud structure, life-time and phase fraction have been removed. At the same time, the continuous treatment of cloud ice with a single category no longer requires the weakly constrained threshold size for the conversion of ice crystals to snow. Together, these factors make the new scheme more closely based on first principles. This reduces its conceptual complexity and simplifies both model development and the interpretation of model results.

An important feature of the original P3 scheme are the rime variables that allow to predict the particle shape and density. We could not produce a purely stratiform cloud formation scenario where rime growth significantly exceeded growth by vapor deposition. The large gap between the resolved scales in ECHAM6-HAM2 and the scales on which hydrometeor collisions take place raises the question to which extent riming can be represented on a physical basis in this framework. The two additional

prognostic variables might be unnecessary for the global model used in this study. To establish the rime fraction and density, up- and downdrafts need to be resolved on the scales of clouds. However, any sub-grid-scale motion in the global model is parameterized. It is therefore indispensable to include an elaborate microphysics scheme in the convection parameterization that is able to represent the co-existence of liquid water and ice. This is not the case for our default scheme. While there have been approaches trying to improve this aspect in the past (Lohmann, 2008; Croft et al., 2012), assessing the rime fraction and density required for the P3 representation of cloud ice requires that the associated cloud parameters would also need to be predicted within the convective parameterization of ECHAM6-HAM2.

We evaluated limitations of the single category scheme. The inability to distinguish particles from different sources, inherent to any bulk particle scheme, persists. At the core of this problem is the fact that it is not the ice particle properties themselves for which prognostic equations are solved but that they are diagnosed from the prognostic ice parameters. A solution has been proposed by Milbrandt and Morrison (2016) by using multiple free categories to give the particle properties themselves a prognostic flavor.

Reducing the number of weakly constrained parameters by going from a multi- to a single category scheme as well as fully resolving the ice formation pathway by the prognostic treatment of cloud ice are clear conceptual improvements over the original scheme. The level of sophistication to which the single category can be implemented in a global model remains to be seen. In the context of climate projections, the benefit from solving additional equations to represent rimed ice properties, as is done in the P3 scheme, or adding multiple free categories need to outweigh the associated computational cost. As a next step we will test the performance of the single category globally.

*Code availability.* The code of the cloud microphysics module (Fortran 95) is available upon request from the corresponding author or as part of the ECHAM6-HAMMOZ chemistry climate model through the HAMMOZ distribution web-page https://redmine.hammoz.ethz.ch/projects/hammoz.

*Competing interests.* The authors declare that they have no conflict of interest.

*Acknowledgements.* This project has been funded by the Swiss National Science Foundation (project number 200021_160177). The authors would like to especially thank an anonymous reviewer for valuable input which led to a significant improvement of the model and manuscript. We thank Hugh Morrison and Jason Milbrandt for sharing their ice particle property lookup tables and corresponding generation code as well as their microphysics code which were of great help to develop our own codes. Furthermore, we thank Jörg Wieder for his work related to the WBF process parameterization. The ECHAM-HAMMOZ model is developed by a consortium composed of ETH Zurich, Max Planck Institut für Meteorologie, Forschungszentrum Jülich, University of Oxford, the Finnish Meteorological Institute and the Leibniz Institute for Tropospheric Research, and managed by the Center for Climate Systems Modeling (C2SM) at ETH Zurich. Special thanks go to Sylvaine Ferrachat for technical support regarding the model.

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

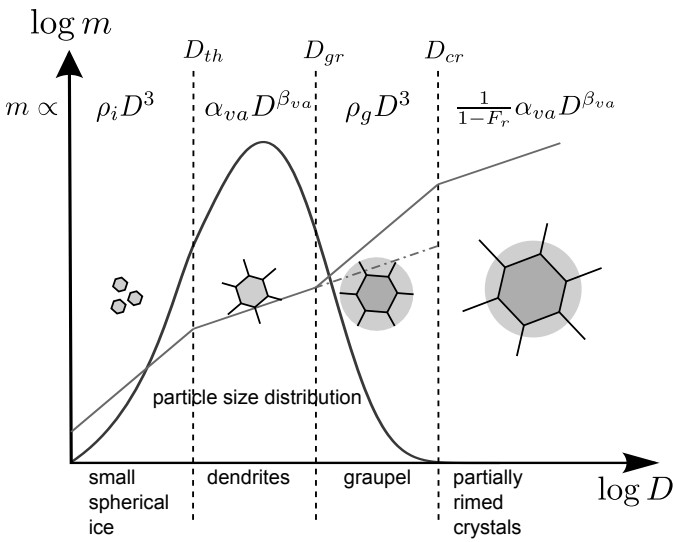

**Figure 1.** Summary of the mechanism used in the single category scheme with an exemplary particle size distribution. $\rho_i$ is the density of ice, $\alpha_{va}$ and $\beta_{va}$ are empirical constants, $\rho_g$ is the diagnosed graupel density and $F_r = q_{rim}/q_i$ is the rime fraction. The dashed-dotted line between $D_{gr}$ and $D_{cr}$ visualizes the extrapolation of the dendritic particles to the rimed regime for the calculation of $\rho_g$ referenced in the text.

|  | *2 category scheme* | *2.5 category scheme* | *1 category scheme* |
|---|---|---|---|
| IC properties | P3 (dendrites) | P3 (dendrites) | P3 |
| IC sedimentation type | off | diagnostic | prognostic |
| IC sedimentation scheme | - | Rotstayn (1997) | upstream Euler |
| Number of prognostic parameters for the ice phase | 2 | 2 | 4 |
| Diagnostic snow | Murakami (1990) | Murakami (1990) | - |

**Table 1.** Summary of the ice description for the three schemes mentioned in the text. Ice crystals are abbreviated by *IC*. P3 represents a description of particle properties according to Sect. 2. Since this analysis is focused on the different cloud ice schemes (prognostic single category vs. diagnostic two category) we use the P3 properties for ice crystals also in the 2 and 2.5 category schemes, assuring comparable fall speeds and deposition rates. Since those schemes only consider riming for snow, P3 is reduced to pure dendritic particles (i.e. $F_r = 0 \Leftrightarrow D_{gr} = D_{cr}$).

Zhang, K., O'Donnell, D., Kazil, J., Stier, P., Kinne, S., Lohmann, U., Ferrachat, S., Croft, B., Quaas, J., Wan, H., Rast, S., and Feichter, J.: The global aerosol-climate model ECHAM-HAM, version 2: sensitivity to improvements in process representations, Atmos. chem. phys., 12, 8911–8949, doi:10.5194/acp-12-8911-2012, 2012.

Zhou, C., Penner, J. E., Lin, G., Liu, X., and Wang, M.: What controls the low ice number concentration in the upper troposphere?, Atmospheric Chemistry and Physics, 16, 12 411–12 424, doi:10.5194/acp-16-12411-2016, 2016.

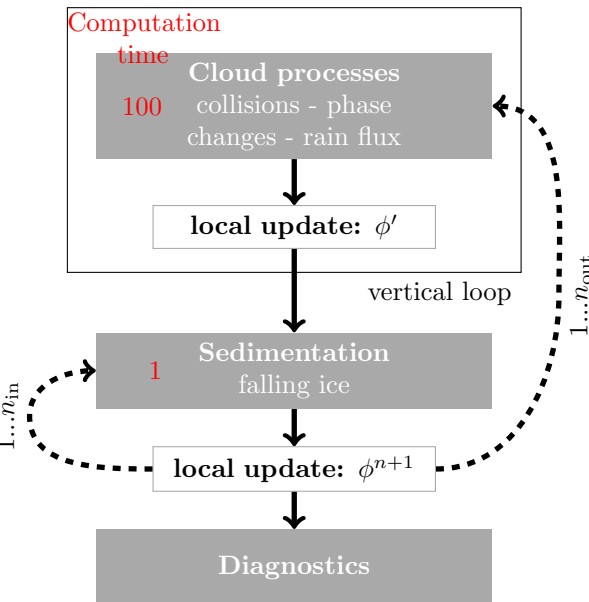

**Figure 2.** Flow diagram of the new scheme. The scheme is divided into 3 major parts: *cloud processes*, *sedimentation* and *diagnostics*. The model state is symbolized by $\phi$ as in the text. Arrows represent the work-flow (solid) and sub-stepping (dashed). The box labeled *vertical loop* represents the part of the code that is looped vertically for the diagnostic treatment of rain. The parameters $n_{\text{in}}$ and $n_{\text{out}}$ refer to the number of iterations of the inner and outer loop respectively. The red numbers on the left represent the approximate computation times of the respective parts in arbitrary units.

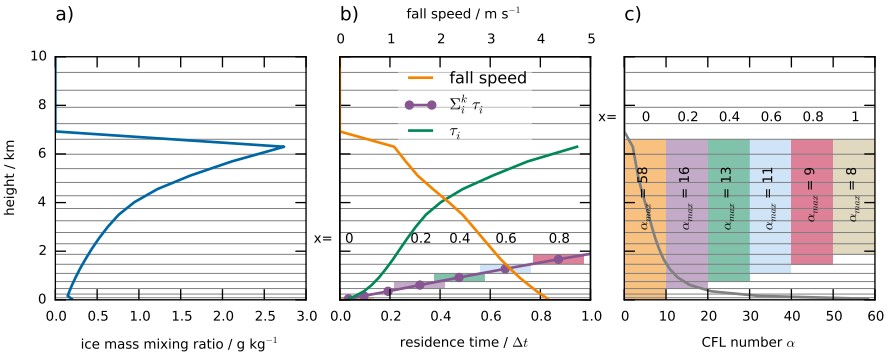

**Figure 3.** Illustration of the optimization approach. Grey horizontal lines show the level interfaces. a) Cloud ice against height, b) fall speed (orange; top axis), residence times (green; bottom axis; per level) and accumulated residence time (purple; bottom axis; accumulated from the bottom). Residence times are given as a fraction of the global time-step $\Delta t$. Colored bars show the levels below which ice only spends a fraction x of the time-step. c) CFL-number $\alpha$. Colored bars show the set of levels $I_{out}(x)$ considered for the calculation of $n_{\text{out}}$.

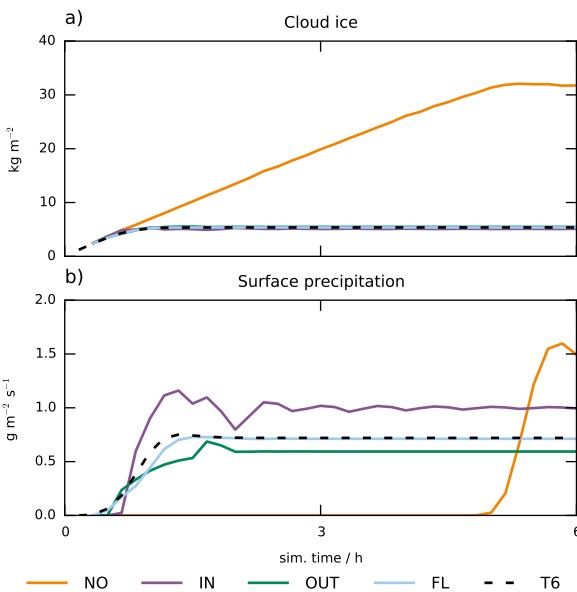

**Figure 4.** Results from a sedimentation test case in the single column model. a) cloud ice and b) surface precipitation. Colors indicate different simulation setups: T6 uses a time-step of $6\,\mathrm{s}$. In the simulations NO, IN, OUT and FL a global time-step of $600\,\mathrm{s}$ is used. They differ in their sub-stepping: FL has full sub-stepping with online computation of $n_{in}$ and $n_{out}$, IN sets $n_{out} = 1$ and only uses the inner loop with online computation of $n_{in}$ and vice versa for the simulation OUT. NO does not use any sub-stepping with constant $n_{in} = n_{out} = 1$.

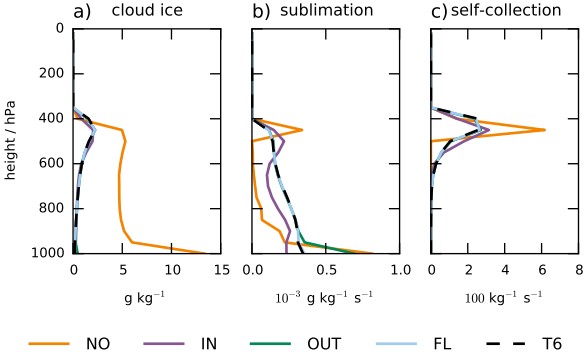

**Figure 5.** Simulations as in Fig. 4 but for vertical profiles at equilibrium, evaluated after $12\,\mathrm{h}$ of the simulation. a) cloud ice, b) sublimation rate and c) self-collection rate.

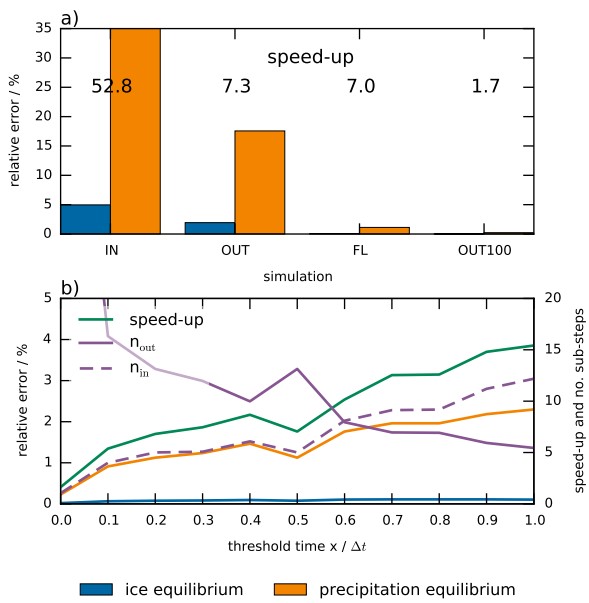

**Figure 6.** Relative errors $|(\phi_{SIM} - \phi_{T6})/\phi_{T6}|$ and speed-up factor $t_{T6}/t_{SIM}$ within the cloud microphysics scheme for the simulations *SIM* shown in Fig. 4 and Fig. 5 together with a new simulation *OUT100* that achieves $\alpha < 1$ throughout the column only by using the outer loop. a) Blue and yellow bars show the relative error of the ice water path and surface precipitation at equilibrium. b) Relative errors as in a) (orange and blue lines). Purple lines show the number of outer (solid; right axis) and inner (dashed; right axis) iterations. The green line shows the speed-up (right axis).

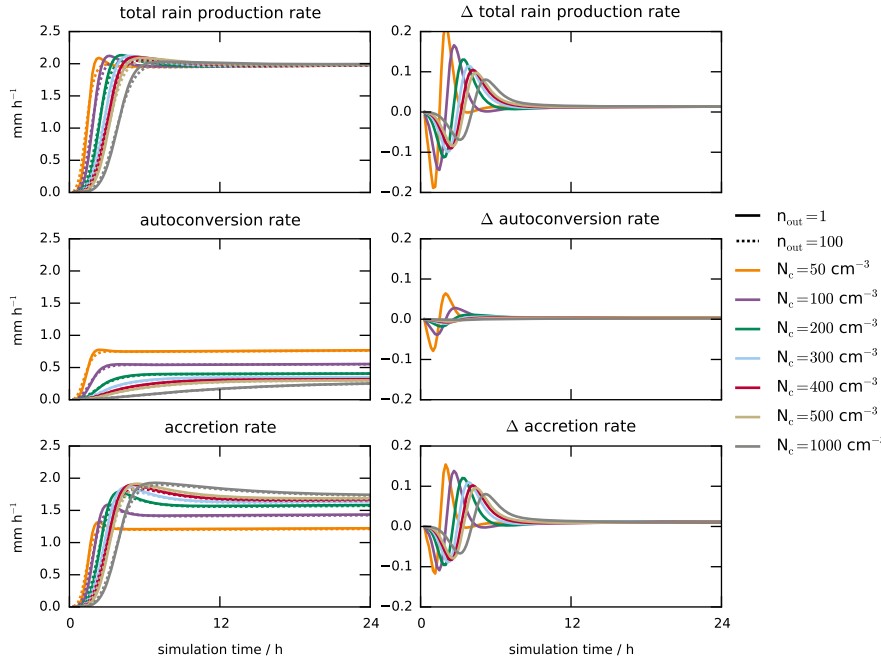

**Figure 7.** Rain production rates for the simulations described in the text. Solid lines are simulations with $n_{out} = 1$ and dashed lines are simulations with $n_{out} = 100$. Different colors represent simulations with different cloud droplet number concentrations reaching from $50\,\mathrm{cm}^{-3}$ to $1000\,\mathrm{cm}^{-3}$. The left column shows the total rain production rate and its constituents autoconversion of cloud droplets to rain drops and accretion of cloud droplets by rain drops. The right column shows the differences ($R^1 - R^{100}$, for $R^n$ being any of the rates above) between the simulations with $n_{out} = 1$ and $n_{out} = 100$ for every process.

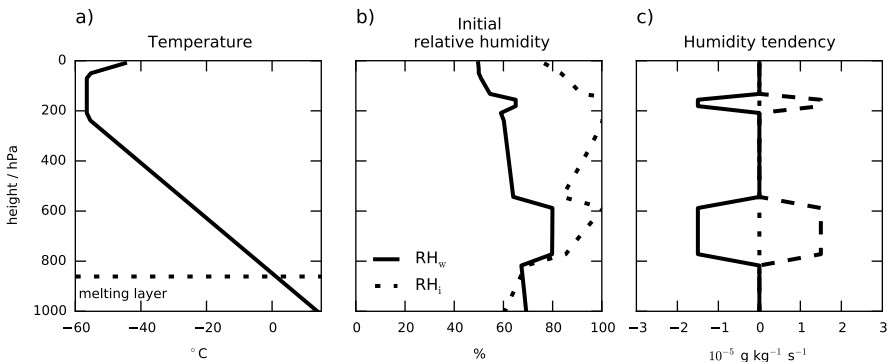

**Figure 8.** Vertical profiles of the single column model initial conditions and forcing terms. a) The temperature is initially set to the international standard atmosphere temperature profile. b) The humidity profile allows for cloudy regions in cirrus and mixed-phase regimes. c) The humidity forcing terms that are applied to initiate cloud formation. Solid lines show the forcing during cloud formation, dotted lines show the stable phase without forcing and the dashed lines show the forcing of cloud dissipation.

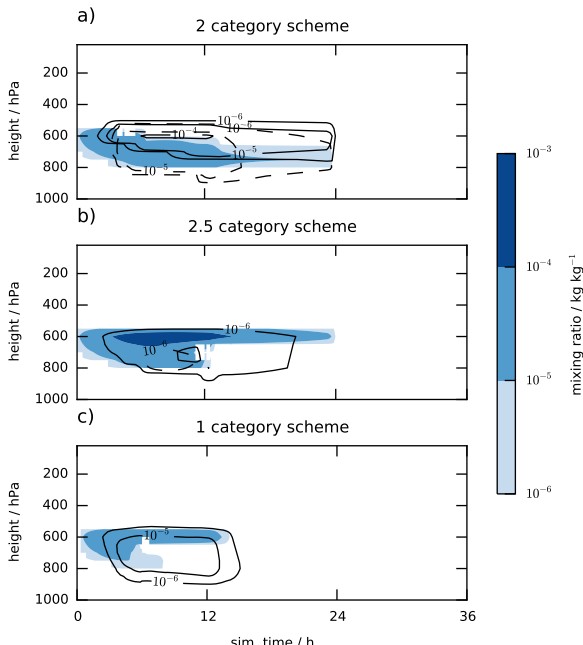

**Figure 9.** Timeseries of the vertical profiles of in-cloud ice (contour lines) and liquid water contents (colors). For the two category schemes snow mass is indicated by dashed lines. Note that snow is a vertically integrated quantity and the profile therefore is only an approximation. The sub-figures (a) and (b) show two versions the original scheme with diagnostic treatment of sedimentation as discussed in the text and sub-figure (c) shows the single category with prognostic ice sedimentation.

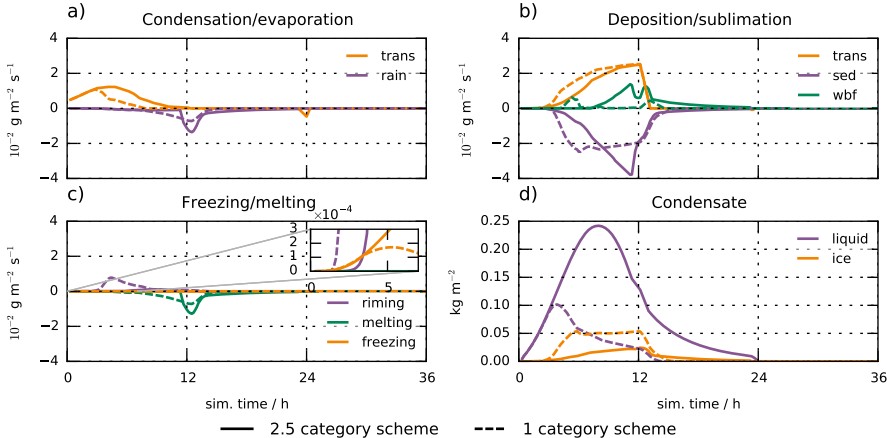

**Figure 10.** Timeseries for the vertically integrated phase change process rates for the simulations described in the text. The solid lines show results from the 2.5 category scheme with diagnostic snow and falling ice and the dashed lines show the results from the single category scheme with prognostic ice sedimentation. The sub-figures (a)-(c) show the rates of phase changes, (d) shows the vertically integrated condensate. Condensation and deposition rates are divided into changes due to by *trans*port (forcing term), evaporation of *rain* and sublimation of *sed*imenting ice. Ice growth at the expense of liquid is denoted by *wbf*.

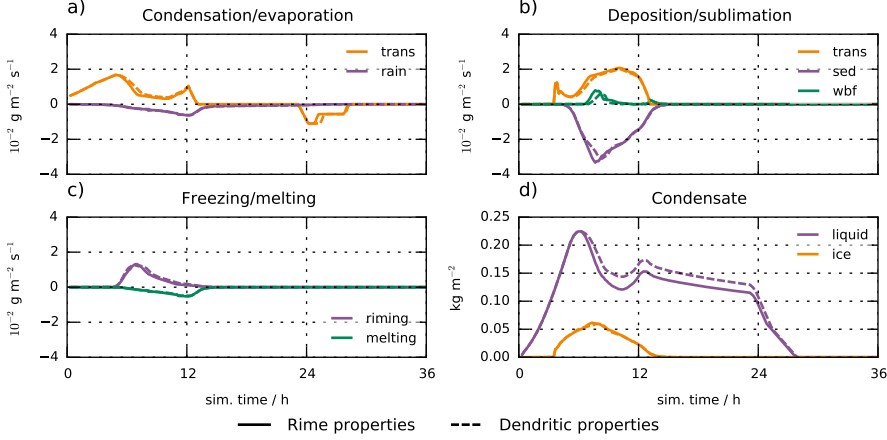

**Figure 11.** Same as in Fig. 10 but for simulations with the single category scheme where freezing in the mixed-phase cloud is turned off and the humidity forcing in the cirrus regime is turned on. The solid lines represent a simulation where riming affects the particle properties and the dashed lines represent a simulation where the mass to size relation for dendrites is assumed for all particles with diameter $D > D_{th}$.

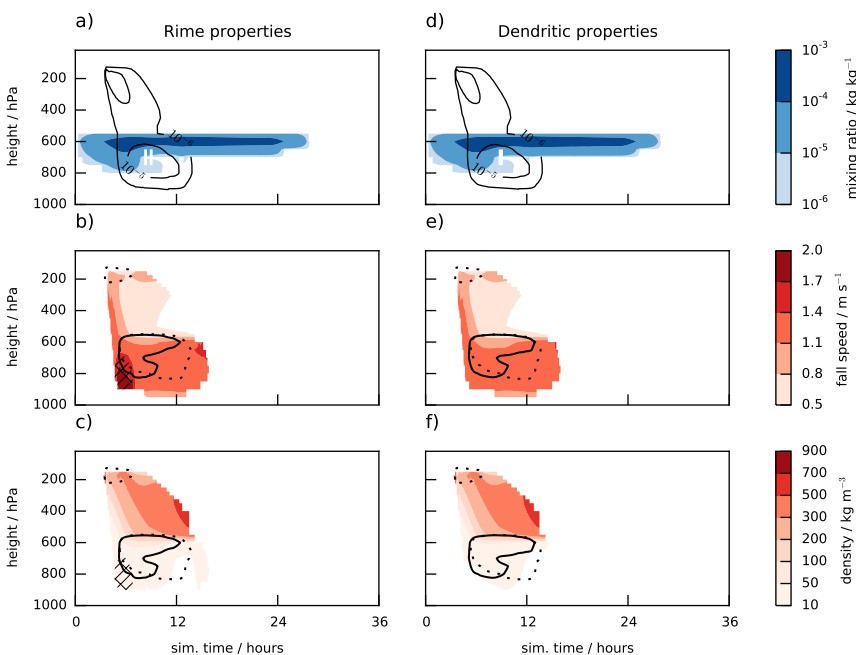

**Figure 12.** Timeseries of the vertical profiles of the particle properties for the seeder-feeder simulation described in the text. Figures a) to c) and d) to f) show results for the simulations with and without taking the change of particle properties due to riming into account respectively. Figures a) and d) show the water mixing ratios (colors for liquid, contours for ice), b) and e) show ice particle fall speeds and c) and f) show ice particle density. In figures b), c), d) and f) the solid and dotted lines show regions of significant ($> 10^{-9}\,\mathrm{kg\,kg^{-1}\,s^{-1}}$) riming and depositional growth respectively. Hatches mark areas where $F_r > 0.5$, i.e. where particles are dominated by rimed ice.

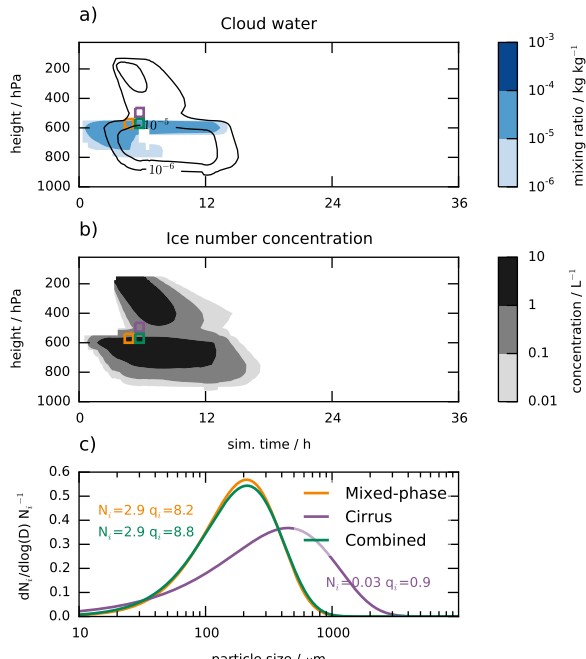

**Figure 13.** Timeseries of the vertical profile of water contents for the cirrus cloud above a mixed-phase cloud simulation described in the text are shown in a) (black solid lines: ice, colors: liquid water). Figure b) shows the same for the ice crystal number concentration. The rectangles in a) and b) visualize the regions representative of the particle size distributions in c) (not drawn to scale). Normalized particle size distributions for the rectangular areas in a) and b) are shown in c). The orange and purple lines represent the mixed-phase and cirrus particles before impact and the green line is the combination of the two after impact. Since the size distribution is normalized we give the total number $N_i$ for each distribution in the corresponding color in $\mathrm{L}^{-1}$ as well as the total ice mass mixing ratio $q_i$ in $\mathrm{\mu g\,kg}^{-1}$