# Peer review of "Prognostic parameterization of cloud ice with a single category in the aerosol-climate model ECHAM(v6.3.0)-HAM(v2.3)"

_Geoscientific Model Development, 2017_

## Short Comment (SC1) · 11 Oct 2017

Remo

As explained in https://www.geoscientific-model-development.net/about/manuscript_ types.html GMD is encouraging that authors upload the program code of models (including relevant data sets) as a supplement or make the code and data available at a data repository preferable with an associated DOI (digital object identifier) for the exact model version described in the paper. If for some reason the code and/or data cannot

be made available in this form authors need to state the reasons for why access is restricted. In your case that would mean that you need to state that use is granted under a special software licensing agreement only which permits the use for research and education free of charge.

All the best Lutz Gross GMD Executive Editor

---

## Author Comment (AC1) · 17 Oct 2017

Dear Lutz Gross

Thank you for your comment concerning code and data accessibility. Indeed, the data used for creating the plots in the manuscript can easily be made accessible and I am happy to share that with the community. I attached all the required data to this comment and will add it as supplement to the final version of the manuscript.

Use of the ECHAM-HAMMOZ software is permitted for research and education. It is

explicitly prohibited to use it or parts of it, whether modified or not, for commercial purposes. Therefore it cannot be made publicly available, which will be stated more precisely in the final version of the manuscript.

Kind regards Remo Dietlicher

Please also note the supplement to this comment:
https://www.geosci-model-dev-discuss.net/gmd-2017-164/gmd-2017-164-AC1-supplement.zip

---

## Referee Comment (RC1) · Anonymous Referee #1 · 26 Oct 2017

This paper describes the implementation of a single ice category into the ECHAM6-HAM2 model. Several modifications were needed to the original microphysical scheme do deal with prognostic sedimentation of the new single ice category. The authors show the impact on including sub-stepping or the sedimentation in the microphysical scheme, based upon previously published studies. Further, they use the single ice category scheme developed by Morrison and Milbrant (2015). The use of a single ice category in both regional and climate models are becoming more tractable. I would have liked to see this scheme tested in a full global run, but that might be too much to

add at this point. So I believe this paper can be published after a few minor changes.

Abstract: Mention new single ice in CAM, but not in the main body. Somewhere in the introduction mention that MM15 is included in CAM.

Page 4, line 1: What is the relation ship between mu and lambda? I suggest giving the equation. Page 4, lines 2-6. Can this sentence be simplified? Perhaps split into two for easier reading.

Page 4, line 14. It seem that Eidhammer et al (2017) also included the single ice category P3 scheme into the global CAM model. I think the fact that other global models have the same single ice category scheme implemented should be mentioned in the introduction and a short discussion on the difference and similarities between the approach in this paper and the one of Eidhammer et al could be included.

Page 8, line 8. What about deposition freezing at cirrus temperatures, and competition between heterogeneous and homogeneous freezing? Is this effect included in the parameterization by Karcher and Lohmann (2002)?

Page 10, line 2. Should it be $S\_acc=d\_qr/dt|acc$ instead of $S\_acc=d\_qr/dt|aut$?

Page 10, line 2: I suggest including "mass" for the ice mixing ratio: ice mass- $q\_i$ .

Page 11, line 16: Is the limit of 0C due to diagnostic rain? I suggest including the reason for the 0C (or actual 5C) limit.

Page 16, line 23. I suggest adding a comma after "…in the cloud"

Page 20, line 9. I suggest renaming the section 5.3 with something more descriptive, since single ice category is considered in all the other sections as well.

Page 20, line 12. Remove comma after "remember"

Page 20, line 14: I suggest reminding the readers of the 4 parameters here. Page 21, line 4: Add "….the number concentration in the tail of the….."

Page 23, line 12. I would like to see a short description in how ECHAM6-HAM2 deals with conversion of ice to snow somewhere in the paper. According to the introduction, Morrison and Gettelman (2008) use a threshold size while, while Murakami (1990) base it on accretion and riming rates. On page 4, it is stated that conversion rates dates back to Murakami (1990), while on page 23, line 12 it is stated that the single ice category scheme removes the threshold size parameter. But does Murakami use a threshold size parameter?

———————————————

---

## Referee Comment (RC2) · Anonymous Referee #2 · 3 Nov 2017

Review of
**Prognostic parameterization of cloud ice with a single category in the aerosol-climate model ECHAM(v6.3.0)-HAM(v2.3)**
by Dietlicher et al.

**General comment:**
In this study a new bulk scheme for representing stratiform cloud microphysics in large-scale models is presented. The model is based on former bulk model schemes with introducing a single prognostic ice category for representing rimed ice particles. The scheme is based on the approach by Morrison and Milbrandt (2015), also called P3 scheme. It is described and some idealized single column model tests are carried out in order to compare the new scheme with former treatment of ice in bulk model schemes for large-scale models.

Although the idea of representing ice in such a consistent way is not completely new, the design of a cloud scheme for large-scale models is an important step towards more consistent and realistic representation of ice, mixed-phase and liquid clouds in large-scale models. Thus, this study is a meaningful contribution for GMD. However, there are several issues, which must be clarified before this manuscript can be accepted for publication. Therefore I recommend major revisions for this manuscript.

In the following I will explain my concerns in details

**Major points**

1. Description of the new scheme
   The description of the treatment of ice particles in a single category is very short and not sufficient for a journal dedicated to model development. Referring to the original publication Morrison and Milbrandt (2015) is not sufficient. Actually, there are several inconsistencies in the text and also between descriptions and figures. For instance, in the text (page 4, lines 9-10) it is stated that depositional growth is assuming spherical shape. However, in figure 1, this is not the case. Thus, the description of the scheme must be extended and inconsistencies in the description must be avoided.

2. Treatment of sedimenting particles

   (a) The treatment of sedimenting particles of different water phases is inconsistent. While sedimenting ice particles are treated prognostically using a time splitting, rain is treated with a diagnostic scheme. Although the authors state that they want to focus on the representation of cloud ice, this is not enough because the P3 scheme actually describes the interaction of liquid and solid cloud particles. Thus, also the treatment of sedimentation should be consistent. Since former work at ETH was carried out on treatment of prognostic rain, it is not really understandable, why the authors restrict the scheme to diagnostic rain.

   (b) For the prognostic treatment of sedimentation of ice a time sub-stepping has been introduced. For the one-dimensional advection in the column an explicit Euler scheme was used. It is not really clear, why an explicit scheme is used, since this has crucial restrictions due to CFL criterion. Why not using an implicit scheme (even of higher order)? Such a scheme would be more robust and the restrictions to the sub time step would be more relaxed, since implicit schemes are commonly more stable.

3. Description of results
   Although the results seem to show an improvement in representation of ice in mixed-phase clouds, the description of the results is a bit confusing and it is hard to follow, what the authors wanted to say. Please state your major results and the improvements due to the introduction of the new scheme in a clearer and more structured way.

**Minor points:**

1. Sub stepping for particle generation?
   It is not clear why sub stepping was not introduced for particle generation, too. Since processes of

activation, freezing or nucleation are highly sensitive to time steps, the existing framework of sub stepping, as designed for other processes, could be used for this purpose. For instance, the resolved dynamics could be used as a criterion, whether particle generation will occur in a time step. Then, particle generation processes could be resolved in the sub stepping. Please comment on this issue.

2. Equation (9) is not consistent with thermodynamics in mixed-phase clouds
In mixed-phase clouds the water vapour is close to equilibrium with respect to liquid phase, i.,e. $RH \sim 1$ until all water has been transformed into ice; then growth of ice particles reduce relative humidity towards ice saturation. Thus, the blend of two different equilibria is not really consistent. Is this quantity only used for cloud cover or is it used for the description of cloud processes in mixed-phase clouds? Please explain this.

3. Equation (11) for growth in WBF process
Is the assumption of planar ice particles consistent with the assumptions in the P3 scheme? Please clarify.

4. Use of TKE scheme for subgrid scale vertical velocity
From a recent study it is known that the TKE based subgrid scale wind parameterisation and the releated ice nucleation significantly overestimates the ice crystal number concentration (Zhou et al., 2016). Please comment, why this parameterisation is used in the model.

5. Section 3.3.4
What is the physical basis for the melting time step of $\tau_{mlt} = 1\,\mathrm{min}$?

6. Autoconversion and accretion parameterisations
In the original article by Khairoutdinov and Kogan (2000) it is stated clearly that their scheme was derived for LES models, i.e. for a spatial resolution of tens of metres. They also stated that the scheme cannot be simply extrapolated for use in large-scale models (see page 231, left column, lines 3-16). Please justify, why this parameterisation is used in a large-scale model with a horizontal resolution of few tens of kilometres.

7. Page 11, lines 21-27:
The description of the simulation scenario, especially of initial and boundary conditions is very short. Please extend the description.

8. Description of figures 3 and 4
Although in the figure a reference simulation FL is indicated, the description of this simulation setup cannot be found in the text. The question arises if there was a series of simulations with decreasing time step leading in convergence to a reference simulation with very short time step. Was FL designed like this? Please explain. The dashed black line in figures 3 and 4 is quite hard to read, please change the line style.

9. Name 2.5 category
Actually, I was a bit confused by the names 1, 2 and 2.5 category. Since in cloud physics often single and double moment schemes are used, and we tend to believe that double moment schemes are better and schemes with more categories are also better, the names are a bit counter-intuitive. Actually, I have no better suggestion; maybe it would help to clarify the names in the very beginning in a more concise way.

**References**

Zhou, C., J. Penner, G. Lin, X. Liu, M. Wang, 2016: What controls the low ice number concentration in the upper troposphere? Atmos. Chem. Phys., 16, 12411-12424, doi:10.5194/acp-16-12411-2016

---

## Author Comment (AC2) · 22 Dec 2017

**Reply to anonymous Referee #1**

**Remo Dietlicher**

**December 22, 2017**

Thank you very much for thoroughly reading our manuscript and pointing out some of the inconsistent descriptions that were present. We are happy about the positive feedback regarding publishing this work. Please find the detailed answers to your comments below.

Abstract: Mention new single ice in CAM, but not in the main body. Somewhere in the introduction mention that MM15 is included in CAM.

It is included in the introduction now.

Page 4, line 1: What is the relation ship between mu and lambda? I suggest giving the equation. Page 4, lines 2-6. Can this sentence be simplified? Perhaps split into two for easier reading.

The chapter describing the P3 scheme has been extended. In doing so, these points have been addressed.

Page 4, line 14. It seem that Eidhammer et al (2017) also included the single ice category P3 scheme into the global CAM model. I think the fact that other global models have the same single ice category scheme implemented should be mentioned in the introduction and a short discussion on the difference and similarities between the approach in this paper and the one of Eidhammer et al could be included.

We agree that it is interesting to compare the implementation of the single ice category in CAM by Eidhammer et al. (2017) to ours in ECHAM-HAM. We added therefore the main differences between the study by Eidhammer et al. (2017) and ours in the introduction.

Page 8, line 8. What about deposition freezing at cirrus temperatures, and competition between heterogeneous and homogeneous freezing? Is this effect included in the parameterization by Karcher and Lohmann (2002)?

The Kärcher and Lohmann (2002) scheme includes homogeneous freezing. There is ongoing work in our group to improve the representation of cirrus cloud formation in ECHAM-HAM, including the competition for vapor deposition between homogeneous and heterogeneous freezing and pre-existing ice crystals. When this is ready, it is planned to be included in our scheme.

Page 10, line 2. Should it be S_acc=d_qr/dt|acc instead of S_acc=d_qr/dt|aut?

Yes, thanks!

Page 10, line 2: I suggest including mass for the ice mixing ratio: ice mass- q_i.

Good idea. This has been done.

Page 11, line 16: Is the limit of 0C due to diagnostic rain? I suggest including the reason for the 0C (or actual 5C) limit.

The way we calculate the number of sub-steps has been reworked to be more general and is now temperature-independent.

Page 16, line 23. I suggest adding a comma after . . .in the cloud

Done.

Page 20, line 9. I suggest renaming the section 5.3 with something more descriptive, since single ice category is considered in all the other sections as well.

It is now called 'Limitations of the P3 scheme'. In fact, we renamed all the subsections of chapter 5 to make the story more clear. It goes from validation (5.1) to adaptation for the GCM (5.2) to limitations (5.3).

Page 20, line 12. Remove comma after remember
Page 20, line 14: I suggest reminding the readers of the 4 parameters here. Page 21, line 4: Add . . ..the number concentration in the tail of the. . ...

Done.

Page 23, line 12. I would like to see a short description in how ECHAM6-HAM2 deals with conversion of ice to snow somewhere in the paper. According to the introduction, Morrison and Gettelman (2008) use a threshold size while, while Murakami (1990) base it on accretion and riming rates. On page 4, it is stated that conversion rates dates back to Murakami (1990), while on page 23, line 12 it is stated that the single ice category scheme removes the threshold size parameter. But does Murakami use a threshold size parameter?

This was indeed not consistent. Murakami (1990) assumes a threshold size for snow which is used together with ice particle growth rates to calculate a characteristic time needed to form snow. This time is then converted to a conversion rate. This is now elaborated more clearly in the introduction.

---

## Author Comment (AC3) · 22 Dec 2017

**Reply to anonymous Referee #2**

**Remo Dietlicher**

**December 22, 2017**

Thank you very much for your valuable input on our scheme. Your suggestions led to significant improvements in terms of accuracy and consistency in the description of ice processes. We removed the separate treatment of in-cloud and grid-box mean processes. This step turned out to be necessary to converge to the high-resolution simulation in an enhanced sedimentation test case. At the same time it allows to make use of the sub-stepping for all the processes. Furthermore, the process rates for melting and deposition are now integrated offline and included in the lookup table to make the ice particle properties and process rates entirely consistent.

For the vertical advection of cloud ice we implemented an implicit Euler scheme as reference. To our understanding, there is no perfect integration method for the vertical advection of cloud ice due to the sharp wave-fronts at cloud base and cloud top. The section 4 on the online computation of the number of iterations of the inner and outer loops has been rewritten to explain the nested sub-stepping method in detail. From this it should be clear that for our purposes, the integration method is only of secondary importance.

The point we could not agree with is whether the diagnostic treatment of rain is inconsistent with a prognostic single category for ice. The only interaction between cloud water and ice that is represented by the original cloud microphysics scheme in ECHAM6-HAM2 is freezing of cloud droplets and riming of snow with cloud droplets. The new scheme does that as well. On top of that, it is able to continuously increase the riming rate with increasing ice particle size due to the single category treatment. We are still missing the interaction between rain and cloud ice and it is questionable whether the large-scale model is able to produce the forcing required to form heavily rimed particles at all. With that in mind, we argue that the new scheme is not less consistent than treating both snow and rain diagnostically.

We agree that the framework established here could be extended to include prognostic rain. However, previous work in our group had a different focus and merging the two is out of the scope of this manuscript but envisioned in future work.

Please find detailed answers to your comments below.

**Major points**

1. Description of the new scheme

The description of the treatment of ice particles in a single category is very short and not sufficient for a journal dedicated to model development. Referring to the original publication Morrison and Milbrandt (2015) is not sufficient. Actually, there are several inconsistencies in the text and also between descriptions and figures. For instance, in the text (page 4, lines 9-10) it is stated that depositional growth is assuming spherical shape.

However, in figure 1, this is not the case. Thus, the description of the scheme must be extended and inconsistencies in the description must be avoided.

We extended the section on the original P3 scheme. It now contains all the information needed to reproduce the P3 lookup tables based on the description in the original P3 paper. As to the inconsistency pointed out here, we changed the wording to better explain what we meant.

2a) The treatment of sedimenting particles of different water phases is inconsistent. While sedimenting ice particles are treated prognostically using a time splitting, rain is treated with a diagnostic scheme. Although the authors state that they want to focus on the representation of cloud ice, this is not enough because the P3 scheme actually describes the interaction of liquid and solid cloud particles. Thus, also the treatment of sedimentation should be consistent. Since former work at ETH was carried out on treatment of prognostic rain, it is not really understandable, why the authors restrict the scheme to diagnostic rain.

We separate this comment into to parts and answer them separately: 1) 'diagnostic rain is inconsistent with the P3 method that explicitely consideres riming' and 2) 'previous work at ETH already involved prognostic rain, why is it not included in this study'.

1) In light of chapter 5.3, we doubt that large-scale models are able to reproduce the conditions which allow for an accurate representation of riming because it relies on the turbulent motion within the cloud. Thus we argue that neglecting riming involving rain drops is not the major concern for a realistic representation of the rime formation.

2) Former work in our group was targeted at the representation of marine stratiform clouds with a focus on cloud droplet activation and the representation of cloud and rain drop spectra. While a completely prognostic scheme is envisioned in future, merging the two is out of the scope of this manuscript.

2b) For the prognostic treatment of sedimentation of ice a time sub-stepping has been introduced. For the one-dimensional advection in the column an explicit Euler scheme was used. It is not really clear, why an explicit scheme is used, since this has crucial restrictions due to CFL criterion. Why not using an implicit scheme (even of higher order)? Such a scheme would be more robust and the restrictions to the sub time step would be more relaxed, since implicit schemes are commonly more stable.

To our understanding, the perfect integration method to solve the advection equation for sedimenting ice does not exist. As it is elaborated more clearly in the text now, the perfect method would need to be non-dispersive, unconditionally stable and able to deal with sharp wave-fronts that are encountered at cloud base and cloud top. While an implicit method satisfies the first two requirements, it does not satisfy the third. At the same time, CFL numbers are not the main concern in our case because ECHAM-HAM uses thin levels close to the ground and broader ones aloft (see the new figure 3). Therefore, the online reduction of the time-step to limit the CFL number in the lowest levels implies that CFL numbers are very small higher up. Since this reduction of the time-step can be done by the very cheap inner loop, the integration method is of secondary importance.

Nevertheless, we implemented a backward Euler method which is still available in the code. All the results using the full sub-stepping shown in the manuscript are almost identical, regardless of the integration method.

3. Description of results

Although the results seem to show an improvement in representation of ice in mixed-phase clouds, the description of the results is a bit confusing and it is hard to follow, what the authors wanted to say. Please state your major results and the improvements due to the introduction of the new scheme in a clearer and more structured way.

We aligned the story around the steps of validation (5.1) to adaptation for the GCM (5.2) to limitations (5.3). For this we renamed the subsection titles and made the text more concise. Especially in section 5.1 we elaborated more precisely how the different microphysics schemes relate to each other and why the new scheme solves many of the problems we had with the previous ones.

**Minor points**

1. Sub stepping for particle generation?

It is not clear why sub stepping was not introduced for particle generation, too. Since processes of activation, freezing or nucleation are highly sensitive to time steps, the existing framework of sub stepping, as designed for other processes, could be used for this purpose. For instance, the resolved dynamics could be used as a criterion, whether particle generation will occur in a time step. Then, particle generation processes could be resolved in the sub stepping. Please comment on this issue.

Now there is sub-stepping for all process rates. Freezing was already part of the sub-stepping loop. Activation of cloud droplets and nucleation of ice crystals in cirrus clouds are parameterized in a somewhat special way. The cirrus and activation methods involve an adiabatic parcel ascent that is assumed to contain the entire process from ascent to particle formation and depletion of supersaturation within a single time-step. This assumption no longer holds for a variable time-step. This is also discussed in the text now.

2. Equation (9) is not consistent with thermodynamics in mixed-phase clouds

In mixed-phase clouds the water vapour is close to equilibrium with respect to liquid phase, i.,e. RH $\sim 1$ until all water has been transformed into ice; then growth of ice particles reduce relative humidity towards ice saturation. Thus, the blend of two different equilibria is not really consistent. Is this quantity only used for cloud cover or is it used for the description of cloud processes in mixed-phase clouds? Please explain this.

Equation (9) is only used for the cloud fraction parametrization, i.e. the cloud fraction $b$ and the water mass that is available for condensation/deposition (or required to evaporate/sublimate) $Q$. The deposition rate is computed using the relative humidity at water saturation within a mixed-phase cloud as long as cloud water is present and the grid-box mean relative humidity otherwise (e.g. for melting and sublimation of cloud ice).

3. Equation (11) for growth in WBF process

Is the assumption of planar ice particles consistent with the assumptions in the P3 scheme? Please clarify.

No and we changed that now. We integrate the process rates for deposition/sublimation and melting (which both depend on the capacitance $C$ and the ventilation coefficient $f_v$) offline. We use different capacities for the different particle property regimes (small spherical ice, dendrites, graupel, partially rimed crystals) to account for the different geometries of the particles.

4. Use of TKE scheme for subgrid scale vertical velocity

From a recent study it is known that the TKE based subgrid scale wind parameterisation and the releated ice nucleation significantly overestimates the ice crystal number concentration (Zhou et al., 2016). Please comment, why this parameterisation is used in the model.

An improved version of the cirrus scheme is being developed in our group. The new scheme will include pre-existing ice crystals, homogeneous and heterogeneous nucleation and a different approach to represent the sub-grid scale updraft. For this reason we use the original parameterization of cirrus clouds in ECHAM6-HAM2. It is interesting to see that for CAM5 the use of TKE leads to an overestimation of the updrafts while in ECHAM5 a study by Joos et. al. 2008 showed that better agreement with observations could be reached when, instead of TKE, gravity waves were used to calculate updraft velocities over mountains.

5. Section 3.3.4

What is the physical basis for the melting time step of mlt = 1 min?

There is none. The goal was to melt all ice within one global model time-step, because this was the assumption in the original scheme. We now included a parameterization based on Mason (1958) [1] found in the book on 'Cloud and precipitation microphysics' by Straka (2009) [2]. Just as with the size dependent deposition rate, this is calculated offline and read back from lookup tables.

6. Autoconversion and accretion parameterisations

In the original article by Khairoutdinov and Kogan (2000) it is stated clearly that their scheme was derived for LES models, i.e. for a spatial resolution of tens of metres. They also stated that the scheme cannot be simply extrapolated for use in large-scale models (see page 231, left column, lines 3-16). Please justify, why this parameterisation is used in a large-scale model with a horizontal resolution of few tens of kilometers.

Yes, that is right. In fact, most parameterizations have been developed either from in-situ data or from process models, both of which are representative for a much smaller scale than a GCM grid-box.

7. Page 11, lines 21-27:

The description of the simulation scenario, especially of initial and boundary conditions is very short. Please extend the description.

The description has been rewritten to include the values for the prescribed tendencies for the four ice moments.

8. Description of figures 3 and 4

Although in the figure a reference simulation FL is indicated, the description of this simulation setup cannot be found in the text. The question arises if there was a series of simulations with decreasing time step leading in convergence to a reference simulation with very short time step. Was FL designed like this? Please explain. The dashed black line in figures 3 and 4 is quite hard to read, please change the line style.

With the major changes to this chapter, the simulations are described more clearly. For us the dashed black line is well readable. Maybe there is an issue with importing figures as pdfs. We will double-check that the final version does not have this problem or maybe switch to a bit-map to assure cross-platform compatability.

9. Name 2.5 category

Actually, I was a bit confused by the names 1, 2 and 2.5 category. Since in cloud physics often single and double moment schemes are used, and we tend to believe that double moment schemes are better and schemes with more categories are also better, the names are a bit counter-intuitive. Actually, I have no better suggestion; maybe it would help to clarify the names in the very beginning in a more concise way.

The paradigm shift that more is not always better when it comes to ice categories is the entire point of the original P3 paper (and to some extent also this work). Therefore it is also inherently counter-intuitive. We tried to highlight the difference between the one and two category schemes by adding a row in table 1 with the number of prognostic parameters and an additional sentence clarifying that the single category actually uses more prognostic parameters than the original ice category.

**References**

[1] B. J. Mason. The physics of clouds. *Q. j. roy. meteor. soc.*, 84(361):304–304, 1958.

[2] J. M. Straka. *Cloud and Precipitation Microphysics: Principles and Parameterizations.* Cambridge University Press, 2009.

---

## Referee Report (RR1)

Review of revised version of
**Prognostic parameterization of cloud ice with a single category in the aerosol-climate model ECHAM(v6.3.0)-HAM(v2.3)**
by Dietlicher et al.

**General comment:**
Almost all comments in the discussion were addressed adequately and the manuscript has improved a lot. However, I have two only minor comments, which should be addressed until the manuscript can be accepted.

**Minor issues:**

1. My comment about the validity of the autoconversion scheme was not really addressed. Of course, parameterisations were developed from different sources. However, if in the original source of the parameterisation the use of the scheme for coarser resolution is seen as questionable (or even as not appropriate), the authors have to state a bit more than their actual answer. Why do they use this parameterisation and what is the actual justification?

2. In principle I agree that there is no best numerical method for treating hyperbolic problems in a conservative and gradient-preserving way. As the authors have pointed out, the integration method is obviously of secondary importance. Nevertheless, the statement that the explicit Euler method is good in terms of gradient-preserving should be omitted; it is very clear that this method is smearing out the gradients in a strong way. Generally, it would have been better to use schemes with flux corrections since they are able to preserve the gradients in a better way. For details see, e.g., LeVeque (2004).

Reference:
LeVeque, R., 2004: Finite Volume Methods for Hyperbolic Problems. Cambridge University Press, Cambridge, UK, 558 p.

---

## Author Response (AR2)

**Reply to anonymous Referee #2 (iteration 2)**

**Remo Dietlicher**

**March 9, 2018**

Thank you for going over the revised manuscript again, especially considering the many changes it contained. I am grateful for your valuable input on model and manuscript!

Please find a point-by-point response below.

**Minor points**

1. My comment about the validity of the autoconversion scheme was not really addressed. Of course, parameterisations were developed from different sources. However, if in the original source of the parameterisation the use of the scheme for coarser resolution is seen as questionable (or even as not appropriate), the authors have to state a bit more than their actual answer. Why do they use this parameterisation and what is the actual justification?

This parameterization is employed in ECHAM-HAM and other global aerosol climate models (Zhang et al., 2016; Section 2, last paragraph) [3]. Justification stems from Wood et al. (2005) [2] who suggest that the scope of the Khairoutdinov and Kogan (2000) [1] autoconversion parameterization may extend beyond its intended use for high-resolution models based on a comparison to autoconversion and accretion rates calculated using observed droplet distributions and the stochastic collection equation.

2. In principle I agree that there is no best numerical method for treating hyperbolic problems in a conservative and gradient-preserving way. As the authors have pointed out, the integration method is obviously of secondary importance. Nevertheless, the statement that the explicit Euler method is good in terms of gradient-preserving should be omitted; it is very clear that this method is smearing out the gradients in a strong way. Generally, it would have been better to use schemes with flux corrections since they are able to preserve the gradients in a better way. For details see, e.g., LeVeque (2004).

That was a bold statement and has now been removed. Thank you for the corresponding literature reference!

[revised manuscript text omitted]